# DYNAMICAL DIFFUSION: LEARNING TEMPORAL DYNAMICS WITH DIFFUSION MODELS

**Xingzhuo Guo**,[*] **Yu Zhang**,[*] **Baixu Chen**,[*] **Haoran Xu, Jianmin Wang, Mingsheng Long**[✉]
School of Software, BNRist, Tsinghua University, Beijing 100084, China
{gxz23,zhangyu24,cbx22,xu-hr22}@mails.tsinghua.edu.cn
{jimwang,mingsheng}@tsinghua.edu.cn

## ABSTRACT

Diffusion models have emerged as powerful generative frameworks by progressively adding noise to data through a forward process and then reversing this process to generate realistic samples. While these models have achieved strong performance across various tasks and modalities, their application to temporal predictive learning remains underexplored. Existing approaches treat predictive learning as a conditional generation problem, but often fail to fully exploit the temporal dynamics inherent in the data, leading to challenges in generating temporally coherent sequences. To address this, we introduce Dynamical Diffusion (DyDiff), a theoretically sound framework that incorporates temporally aware forward and reverse processes. Dynamical Diffusion explicitly models temporal transitions at each diffusion step, establishing dependencies on preceding states to better capture temporal dynamics. Through the reparameterization trick, Dynamical Diffusion achieves efficient training and inference similar to any standard diffusion model. Extensive experiments across scientific spatiotemporal forecasting, video prediction, and time series forecasting demonstrate that Dynamical Diffusion consistently improves performance in temporal predictive tasks, filling a crucial gap in existing methodologies. Code is available at this repository: https://github.com/thuml/dynamical-diffusion.

## 1 INTRODUCTION

Diffusion models (Sohl-Dickstein et al., 2015; Song & Ermon, 2019; Ho et al., 2020; Song et al., 2020) refer to a class of generative models that progressively corrupt data by adding noise through a "forward process" and then iteratively denoise a random input during inference to generate highly realistic samples via the "reverse process". This unique approach has positioned them as a powerful alternative to traditional generative methods. To date, diffusion models have demonstrated strong performance across a wide range of tasks (Kingma et al., 2021; Saharia et al., 2022a;b; Dhariwal & Nichol, 2021) and data modalities (Kong et al., 2021; Chen et al., 2021; Yang et al., 2023; Ho et al., 2022; Blattmann et al., 2023a).

Due to their strong capability to model data distributions, diffusion models are gaining attention in the field of temporal predictive learning. Several recent studies (Voleti et al., 2022; Gao et al., 2023) have explored the application of diffusion models to predictive learning tasks by reinterpreting these tasks as conditional generation problems. In this approach, the model is trained to predict the future conditioned on historical data, such as predicting the next video frame based on preceding frames. Despite yielding promising results, these methods did not explicitly leverage the temporal nature of the data, which may pose challenges for generating temporally coherent sequences (Blattmann et al., 2023a;b). While increasing the capacity of deep models can alleviate this issue, *the fundamental challenge of integrating temporal dynamics into diffusion processes remains underexplored*.

To this end, we propose Dynamical Diffusion (DyDiff), a framework that defines temporally aware forward and reverse diffusion processes. Specifically, in the forward process, each latent is not only modified through the conventional noise addition procedure but is also derived from its temporally

---

[*]Equal contribution.

preceding latent. In this way, Dynamical Diffusion explicitly captures temporal transitions at each diffusion step. Through a theoretical derivation, we establish the existence of the corresponding reverse process and extend it to generate multi-step predictions simultaneously. By leveraging the reparameterization trick, the learning of Dynamical Diffusion is formulated into feasible optimization objectives. This enables efficient training with no additional computational cost compared to standard diffusion models and facilitates efficient sampling similar to the standard DDPM (Ho et al., 2020) and its variants (Song et al., 2021).

Our contributions can be summarized as follows:

- We investigate temporal predictive learning using diffusion models and highlight the underexplored challenge of integrating temporal dynamics into the diffusion process.

- We introduce Dynamical Diffusion, a theoretically guaranteed framework that explicitly models temporal transitions at each diffusion step. We outline key design choices that enable efficient training and inference of Dynamical Diffusion.

- We conduct experiments on various tasks across different modalities, including scientific spatiotemporal forecasting, video prediction, and time series forecasting. The results demonstrate that the proposed Dynamical Diffusion framework consistently enhances performance in predictive learning.

## 2 PRELIMINARIES

**Diffusion models.**   Diffusion models (Sohl-Dickstein et al., 2015; Ho et al., 2020) and their variants (Song & Ermon, 2019; Song et al., 2020) have shown outstanding capabilities in capturing complex data distributions. The core design of diffusion models involves dual forward and reverse processes. Formally, the forward process gradually corrupts real data $\mathbf{x}_0 \sim q(\mathbf{x}_0)$ according to a noise schedule $\{\bar{\alpha}_t\}_{t=1}^T$. At timestep $t$, the corrupted data $\mathbf{x}_t$ can be sampled as

$$\mathbf{x}_t = \sqrt{\bar{\alpha}_t}\mathbf{x}_0 + \sqrt{1 - \bar{\alpha}_t}\boldsymbol{\epsilon}_t, \tag{1}$$

where $\boldsymbol{\epsilon}_t \sim \mathcal{N}(\mathbf{0}, \mathbf{I})$ denotes a random Gaussian noise. Subsequently, in the reverse process, a nerual network $\boldsymbol{\epsilon}_\theta$ is trained to invert forward process corruptions with $p_\theta\left(\mathbf{x}_{t-1}|\mathbf{x}_t\right)$, with the objective of minimizing the variational lower bound

$$L(\theta) = \mathbb{E}_{t,\mathbf{x}_0,\boldsymbol{\epsilon}_t \sim \mathcal{N}(\mathbf{0},\mathbf{I})} \left[ \left\| \boldsymbol{\epsilon}_t - \boldsymbol{\epsilon}_\theta \left( \sqrt{\bar{\alpha}_t}\mathbf{x}_0 + \sqrt{1 - \bar{\alpha}_t}\boldsymbol{\epsilon}_t, t \right) \right\|^2 \right]. \tag{2}$$

Once trained, sampling in diffusion models is performed by iterative denoising from $\mathbf{x}_T \sim \mathcal{N}(\mathbf{0}, \mathbf{I})$ to $\mathbf{x}_0$. Similar to other types of generative models, diffusion models are in principle capable of modeling conditional distributions. This can be achieved by modifying the reverse process to learn $p_\theta(\mathbf{x}_{t-1}|\mathbf{x}_t, \mathbf{c})$, where $\mathbf{c}$ represents the condition.

**Predictive learning with diffusion models.**   The goal of predictive learning is to predict future states $\mathbf{x}^{1:S}$ based on observations $\mathbf{x}^{-P:0}$. By substituting the condition $\mathbf{c}$ with observations $\mathbf{x}^{-P:0}$, the predictive learning task can be naturally interpreted as a conditional generation task, making it well-suited for diffusion models to solve. This approach requires minimal modifications to the original diffusion process and has been adopted by several recent works (Voleti et al., 2022; Gao et al., 2023).

## 3 METHOD

We observe that, when integrating diffusion models into predictive learning, there are two notable axes along which the model must learn simultaneously. The first axis, referred to as the *"prediction axis"*, requires the model to learn the temporal dynamics of the data. The second axis, termed the *"denoising axis"*, necessitates that the model distinguishes noise from corrupted states.

From this perspective, we identify a mismatch in previous methodologies. As shown in Figure 1a, the forward process in standard diffusion models progresses solely along the denoising axis. In particular, historical observations $\mathbf{x}_0^{-P:0}$ serve only as conditions for denoising networks, with no

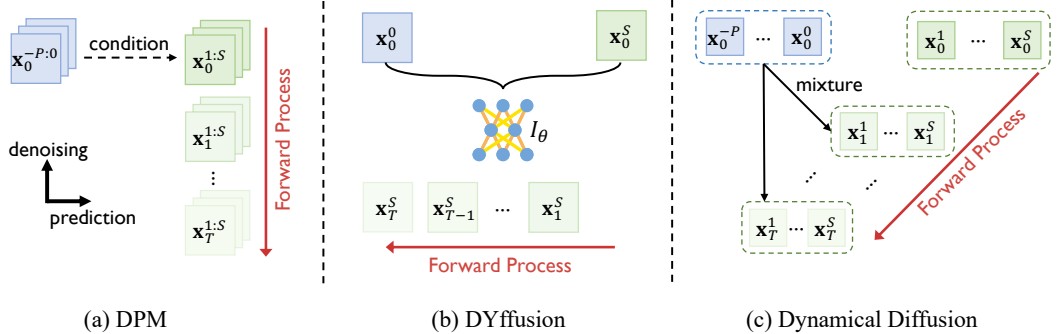

Figure 1: Comparison of diffusion modeling approaches in predictive learning.

temporal dependency considered between temporally adjacent latents $\mathbf{x}_t^s$ and $\mathbf{x}_t^{s-1}$. This modeling strategy isolates the predictive task along the denoising axis, while overlooking the internal continuity and forecasting capabilities of the dynamics that could potentially enhance the diffusion process.

In contrast, an alternative process in (Rühling Cachay et al., 2023), as depicted in Figure 1b, progressively generates intermediates between two states. The forward and reverse processes are modeled as temporal interpolation and extrapolation, respectively. In this approach, the diffusion process is applied on the prediction axis. While this addresses the mismatch issue, it requires the predictability of the intermediate states. Furthermore, the ability to generate high-quality samples has not been fully validated, as the core mechanism of adding and removing noise in DPMs has been eliminated.

Building on the aforementioned considerations, we propose Dynamical Diffusion (DyDiff), designed to concurrently model both the denoising and prediction axes. As illustrated in Figure 1c, Dynamical Diffusion explicitly introduces the mixture of historical states in the diffusion process. By controlling different mixing manners with respect to the timestep $t$, Dynamical Diffusion enables temporal-aware forward and reverse processes, which we present in Subsections 3.1 and 3.2, respectively.

## 3.1 FORWARD PROCESS

In the standard forward process, the corrupted latent at diffusion step $t$ is constructed as $\mathbf{x}_t = \sqrt{\bar{\alpha}_t}\mathbf{x}_0 + \sqrt{1 - \bar{\alpha}_t}\boldsymbol{\epsilon}_t$. Inspired by recurrent neural networks (Hochreiter, 1997; Chung et al., 2014) and state-space models (Gu et al., 2022) that capture temporal transitions through iterative structures, in Dynamical Diffusion, we define each latent $\mathbf{x}_t^s$ by the combination with its previous latent $\mathbf{x}_t^{s-1}$, formalized as follows:

$$\mathbf{x}_t^s = \sqrt{\bar{\gamma}_t} \cdot \left( \sqrt{\bar{\alpha}_t}\mathbf{x}_0^s + \sqrt{1 - \bar{\alpha}_t}\boldsymbol{\epsilon}_t^s \right) + \sqrt{1 - \bar{\gamma}_t} \cdot \mathbf{x}_t^{s-1}, \tag{3}$$

where $\{\bar{\gamma}_t\}_{t=1}^T$ are the newly introduced timestep-aware schedule hyperparameters to control the dependence of the previous latent. By expanding the above equation along the prediction axis, we obtain

$$\mathbf{x}_t^s = \sqrt{\bar{\alpha}_t} \cdot \texttt{Dynamics}\left( \mathbf{x}_0^{-P:s}; \bar{\gamma}_t \right) + \sqrt{1 - \bar{\alpha}_t} \cdot \widetilde{\boldsymbol{\epsilon}}_t^s, \tag{4}$$

where

$$\texttt{Dynamics}\left( \mathbf{x}_0^{-P:s}, \bar{\gamma}_t \right) = \sqrt{\bar{\gamma}_t} \cdot \mathbf{x}_0^s + \sqrt{1 - \bar{\gamma}_t} \cdot \texttt{Dynamics}\left( \mathbf{x}_0^{-P:s-1}, \bar{\gamma}_t \right), \tag{5}$$

and

$$\widetilde{\boldsymbol{\epsilon}}_t^s = \sqrt{\bar{\gamma}_t} \cdot \boldsymbol{\epsilon}_t^s + \sqrt{1 - \bar{\gamma}_t} \cdot \widetilde{\boldsymbol{\epsilon}}_t^{s-1} \sim \mathcal{N}(\mathbf{0}, \mathbf{I}) \tag{6}$$

represents non-independent random Gaussian noise (proof in Appendix A.1). The definition of Dynamics refers to a timestep-aware mixture of all historical states, and further achieves temporal dynamics by adequately controlling the factor $\bar{\gamma}_t$. Notably, the noise factor $\sqrt{1 - \bar{\alpha}_t}$ remains unchanged regardless of the choice of $\bar{\gamma}_t$ and is identical to that in the standard diffusion process. As a result, the signal-to-noise ratio (SNR) in the diffusion model is preserved.

**Discussion on** $\bar{\gamma}_t$  Based on Equation (5), when $\bar{\gamma}_t \to 1$, all prior information is ignored, and the forward process approximates the one in standard diffusion models. Conversely, as $\bar{\gamma}_t$ decreases, earlier historical observations are given greater weight. In designing the schedule for $\{\bar{\gamma}_t\}_{t=1}^T$, it is advisable for $\bar{\gamma}_t$ to be a non-increasing function, ensuring that larger values of $t$ correspond to a stronger emphasis on historical states. Additionally, setting $\bar{\gamma}_0 = 1$ guarantees that $\mathbf{x}_0^s = \mathbf{x}^s$, preserving the initial state. Notably, unlike $\bar{\alpha}_T \approx 0$ in standard diffusion models, it is not necessary for $\bar{\gamma}_T \approx 0$. If $\bar{\gamma}_T$ approaches zero, the reverse process would start from $\texttt{Dynamics}(\mathbf{x}_0^{-P:s}) = \mathbf{x}_0^P$ for all $s$, which might be less relevant than utilizing recent states. In practice, we adopt the schedule $\bar{\gamma}_t = \eta\bar{\alpha}_t + (1 - \eta)$, where $\eta \in [0, 1]$ is a time-independent factor. By default, we set $\eta = 0.5$. We will further analyze the effect of $\eta$ in Section 4.4.

## 3.2  REVERSE PROCESS

The forward process defines a marginal distribution $q(\mathbf{x}_t^s|\mathbf{x}_0^{-P:s})$ that is composed of both random noises and historical states. Next, we discuss the posterior distribution and the reverse process for Dynamical Diffusion.

### 3.2.1  SINGLE-STEP PREDICTION CASE

We begin with the single-step prediction case, i.e. $S = 1$. In this scenario, all the previous states $\mathbf{x}_0^{-P:S-1}$ involved in the diffusion process are fully known. We formulate the following theorems, with proof attached in Appendix A.2.

**Theorem 1.** In a manner akin to DDIM (Song et al., 2021), there exists a non-Markovian forward process with the following marginal distribution

$$q\left(\mathbf{x}_t^1|\mathbf{x}_0^{-P:1}\right) = \mathcal{N}(\sqrt{\bar{\alpha}_t} \cdot \texttt{Dynamics}(\mathbf{x}_0^{-P:1}; \bar{\gamma}_t), (1 - \bar{\alpha}_t)\mathbf{I}). \tag{7}$$

Furthermore, learning of the reverse process can be reparameterized into the following denoising objective

$$L(\theta) = \mathbb{E}_{t,\mathbf{x}_0^{-P:1},\boldsymbol{\epsilon}_t \sim \mathcal{N}(\mathbf{0},\mathbf{I})}\left[\left\|\boldsymbol{\epsilon}_t - \boldsymbol{\epsilon}_\theta\left(\sqrt{\bar{\alpha}_t} \cdot \texttt{Dynamics}(\mathbf{x}_0^{-P:1}; \bar{\gamma}_t) + \sqrt{1 - \alpha_t} \cdot \boldsymbol{\epsilon_t}, t\right)\right\|^2\right], \tag{8}$$

with a DDIM-like sampler

$$p_\theta(\mathbf{x}_{t-1}^1|\mathbf{x}_t^1, \mathbf{x}_0^{-P:0}) = \mathcal{N}\left(\sqrt{\bar{\alpha}_{t-1}} \cdot \texttt{Dynamics}(\mathbf{x}_0^{-P:0}, \mathbf{x}_{\text{pred}}^1; \bar{\gamma}_{t-1}) + \sqrt{1 - \bar{\alpha}_{t-1} - \sigma_t^2}\boldsymbol{\epsilon}_\theta, \sigma_t^2\mathbf{I}\right) \tag{9}$$

where

$$\mathbf{x}_{\text{pred}}^1 = \left(\frac{\mathbf{x}_t^1 - \sqrt{1 - \bar{\alpha}_t}\boldsymbol{\epsilon}_\theta(\mathbf{x}_t^1, \mathbf{x}_0^{-P:0}, t)}{\sqrt{\bar{\alpha}_t}} - \sqrt{1 - \bar{\gamma}_t} \cdot \texttt{Dynamics}\left(\mathbf{x}_0^{-P:0}, \bar{\gamma}_t\right)\right) / \sqrt{\bar{\gamma}_t} \tag{10}$$

refers to the predicted ground truth.

**Theorem 2.** (Informal) There exists a DDPM-like (Ho et al., 2020) Markovian forward process which shares the same marginal distribution as Equation (7), and the reverse process can be learned using the same objective function as Equation (8) and inferred using a DDPM-like sampler

$$p_\theta(\mathbf{x}_{t-1}^1|\mathbf{x}_t^1, \mathbf{x}_0^{-P:0}) = \mathcal{N}(\widetilde{\boldsymbol{\mu}}_t(\mathbf{x}_t^1, \mathbf{x}_{\text{pred}}^1, \mathbf{x}_0^{-P:0}), \sigma_t^2\mathbf{I}) \tag{11}$$

with $\widetilde{\boldsymbol{\mu}}_t$ referring to the posterior mean derived by the forward process.

**Remarks.**  Theorems 1,2 indicate that when $S = 1$, it is feasible to learn a denoiser network similar to standard diffusion models. This denoiser serves as a reparameterization of the reverse process and minimizes the variational lower bound on the posterior distribution. The main difference is that the denoiser in Dynamical Diffusion aims to distinguish from the noisy disturbance of $\texttt{Dynamics}(\mathbf{x}_0^{-P:1})$ instead of $\mathbf{x}_0^1$.

### 3.2.2 Extention to Multi-step Prediction

We now extend the proposed reverse process to the multi-step prediction scenario, i.e., $S > 1$. Compared with the case when $S = 1$, the forward process additionally introduces dependencies among multiple latents, and the reverse process must consider the absence of previous ground truth $\mathbf{x}_0^{1:s-1}$ for a given $s$. The following theorem presents the reparameterized objective, with detailed proof in Appendix A.3.

**Theorem 3.** (Informal) There exists a DDIM-like and a DDPM-like forward process satisfying the marginal distribution

$$q\left(\mathbf{x}_t^{1:S}|\mathbf{x}_0^{-P:S}\right) = \mathcal{N}(\sqrt{\bar{\alpha}_t} \cdot \mathtt{Dynamics}(\mathbf{x}_0^{-P:S}; \bar{\gamma}_t), (1 - \bar{\alpha}_t)\mathbf{J}_t), \tag{12}$$

where $\mathbf{J}_t$ is a non-identity covariance matrix with $(\mathbf{J}_t)_{ik} = (\sqrt{\bar{\gamma}_t})^{i-k}$. Additionally, the reverse process can be reparameterized into the following denoising objective

$$L(\theta) = \mathbb{E}_{t,\mathbf{x}_0^{-P:S}, \widetilde{\epsilon}_t \sim \mathcal{N}(\mathbf{0}, \mathbf{J}_t)} \left[\left\|\widetilde{\epsilon}_t - \epsilon_\theta\left(\sqrt{\bar{\alpha}_t} \cdot \mathtt{Dynamics}(\mathbf{x}_0^{-P:S}) + \sqrt{1-\bar{\alpha}_t} \cdot \widetilde{\epsilon}_t, t\right)\right\|^2\right] \tag{13}$$

with DDIM/DDPM-like samplers which are extensions of Equations (9) and (11).

**Remarks.** Equation (12) naturally generalizes the case when $S = 1$. Specifically, for each state $s$, the marginal distribution $q(\mathbf{x}_t^s|\mathbf{x}_0^{-P:s})$ retains exactly the same form as Equation (7). When considering all latents $\mathbf{x}_t^s$ collectively as a joint distribution, Dynamical Diffusion differs from standard diffusion models in both the forward and reverse processes.

- In the forward process, as discussed in Equation (3), the latents are dependently defined. This dependency leads to a non-identity covariance matrix when combining all states into a joint distribution. Consequently, the denoiser must learn from non-independent sampled noises $\widetilde{\epsilon}$, accommodating the correlations introduced by the dependent states.

- In the reverse process, when reconstructing $\mathbf{x}_{\mathrm{pred}}^{1:S}$ from the current latents $\mathbf{x}_t^{1:S}$ and the predicted noises $\widetilde{\epsilon}_t^{1:S}$, the addition of noises to the dynamics necessitates computing the inverse dynamics function (see Appendix B). The sampler must then reapply noises to the recalculated dynamics to accurately recover the predicted states. Similar algorithms are employed on $\widetilde{\epsilon}_t^{1:S}$ to obtain $\epsilon_t^{1:S}$, ensuring consistency in the reverse diffusion steps.

**Algorithm.** The pseudocode for the training and inference processes of Dynamical Diffusion is provided in Algorithms 1 and 2, respectively. Compared with standard diffusion models, Dynamical Diffusion differs only in its preparation of inputs and outputs for the denoiser $\epsilon_\theta$, without introducing any additional forward or backward passes. Consequently, the computational cost remains similar to that of standard approaches.

## 4 Experiments

In this section, we evaluate Dynamical Diffusion (DyDiff) in three different settings and compare its performance against the standard diffusion model (DPM). We demonstrate that DyDiff is versatile to provide competitive performance across a range of tasks (Section 4.1, 4.2, and 4.3) and conduct in-depth analysis to understand the prediction process of DyDiff (Section 4.4). Unless specifically mentioned, we use the framework of Stable Video Diffusion (Blattmann et al., 2023a), which achieves the state-of-the-art performance on video generation tasks. We provide experimental details in Appendix C, along with additional comparisons presented in Appendix D.

### 4.1 Scientific Spatiotemporal Forecasting

**Setup.** We begin by evaluating the models' performance in scientific spatiotemporal forecasting using the Turbulence Flow dataset (Wang et al., 2020) and the SEVIR dataset (Veillette et al., 2020). This scenario requires the model to learn the underlying physical dynamics. Turbulence Flow is a simulated dataset governed by partial differential equations (PDEs), capturing spatiotemporal dynamics of turbulent fluid flows, specifically the velocity fields. Each frame contains two channels

**Algorithm 1** Training of Dynamical Diffusion

1:  **procedure** Dynamics($\mathbf{x}_0^{L:R}, \bar{\gamma}$)
2:      $\mathbf{x}_{\text{dyn}}^L \leftarrow \mathbf{x}_0^L$
3:      **for** $s$ in $[L+1, R]$ **do**
4:          $\mathbf{x}_{\text{dyn}}^s \leftarrow \sqrt{1 - \bar{\gamma}}\mathbf{x}_{\text{dyn}}^{s-1} + \sqrt{\bar{\gamma}}\mathbf{x}_0^s$
5:      **end for**
6:      **return** $\mathbf{x}_{\text{dyn}}^{L:R}$
7:  **end procedure**
8:
9:  **while** not converged **do**
10:      Sample $\mathbf{x}^{-P:S} \sim \mathcal{X}$
11:      Sample $\boldsymbol{\epsilon}^{1:S} \sim \mathcal{N}(\mathbf{0}, \mathbf{I})$, $t \sim \mathcal{U}[1, T]$
12:      $\mathbf{x}_{\text{dyn}}^{1:S} \leftarrow$ Dynamics($\mathbf{x}_0^{-P:S}, \bar{\gamma}_t$)$^{1:S}$
13:      $\boldsymbol{\epsilon}_{\text{dyn}}^{1:S} \leftarrow$ Dynamics($\boldsymbol{\epsilon}^{1:S}, \bar{\gamma}_t$)
14:      $L(\theta) \leftarrow \big[\big\|\boldsymbol{\epsilon}_{\text{dyn}}^{1:S} - \boldsymbol{\epsilon}_\theta(\sqrt{\bar{\alpha}_t}\mathbf{x}_{\text{dyn}}^{1:S}$
            $+ \sqrt{1 - \bar{\alpha}_t}\boldsymbol{\epsilon}_{\text{dyn}}^{1:S}, \mathbf{x}_0^{-P:0}, t)\big\|^2\big]$
15:      Backprop with $L(\theta)$ and update $\theta$
16:  **end while**
17:  **return** $\theta$

**Algorithm 2** Inference of Dynamical Diffusion

**Require:**
    **procedure** InverseDynamics (Algorithm 3)

1:  Sample $\mathbf{x}_{\text{pred}}^{1:S} \sim \mathcal{N}(\mathbf{0}, \mathbf{I})$
2:  $\mathbf{x}_T^{1:S} \leftarrow$ Dynamics($\mathbf{x}_t^{1:S}, \bar{\gamma}_t$)
3:
4:  **for** $t$ in $[T, 1]$ **do**
5:      $\boldsymbol{\epsilon}_t^{1:S} \leftarrow \boldsymbol{\epsilon}_\theta(\mathbf{x}_t^{1:S}, \mathbf{x}_0^{-P:0}, t)$
6:      $\mathbf{x}_{\text{dyn}}^{1:S} \leftarrow \left(\mathbf{x}_t^{1:S} - \sqrt{1 - \bar{\alpha}_t}\boldsymbol{\epsilon}_t^{1:S}\right) / \sqrt{\bar{\alpha}_t}$
7:      $\mathbf{x}_{\text{dyn}}^{-P:0} \leftarrow$ Dynamics($\mathbf{x}_0^{-P:0}, \bar{\gamma}_t$)
8:      $\mathbf{x}_{\text{pred}}^{-P:S} \leftarrow$ InverseDynamics($\mathbf{x}_{\text{dyn}}^{-P:S}, \bar{\gamma}_t$)
9:      $\boldsymbol{\epsilon}_{\text{pred}}^{1:S} \leftarrow$ InverseDynamics($\boldsymbol{\epsilon}_t^{1:S}, \bar{\gamma}_t$)
10:     $\boldsymbol{\epsilon}_{t-1}^{1:S} \leftarrow$ Dynamics($\boldsymbol{\epsilon}_{\text{pred}}^{1:S}, \bar{\gamma}_{t-1}$)
11:     $\mathbf{x}_{t-1}^{1:S} \leftarrow \sqrt{\bar{\alpha}_{t-1}} \cdot$ Dynamics($\mathbf{x}_{\text{pred}}^{-P:S}, \bar{\gamma}_{t-1}$)$^{1:S}$
            $+ \sqrt{1 - \bar{\alpha}_{t-1}}\boldsymbol{\epsilon}_{t-1}^{1:S}$
12:  **end for**
13:  **return** $\mathbf{x}_0^{1:S}$

representing turbulent flow velocity along the $x$ and $y$ directions. The task is to predict future velocity fields based on prior observations. Following the configuration of Wang et al., we generate sequences of 15 frames at a spatial resolution of $64 \times 64$ grids, using 4 input frames to predict the subsequent 11 frames. SEVIR is a large-scale dataset curated specifically for meteorology and weather forecasting research. Each sample in SEVIR represents $384\text{km} \times 384\text{km}$ observation sequences over 4 hours. Following (Gao et al., 2023), we select the task of predicting Vertically Integrated Liquid (VIL), where the model learns to forecast future precipitation levels. For this dataset, 7 input frames are used to predict the next 6 frames, with each frame having a resolution of $128 \times 128$ grids.

For evaluation, we report the neighborhood-based Continuous Ranked Probability Score (CRPS) (Gneiting & Raftery, 2007) and Critical Success Index (CSI) (Schaefer, 1990; Jolliffe & Stephenson, 2012), following (Ravuri et al., 2021; Zhang et al., 2023). The CRPS metric emphasizes the model's ensemble forecasting capabilities, while the CSI metric evaluates the accuracy of the model's predictions in peak regions. Lower CRPS values and higher CSI scores indicate better performance.

Table 1: Scientific spatiotemporal forecasting results on the SEVIR and Turbulence Flow datasets. $w$, $avg$, and $max$ represent hyperparameters in evaluation metrics (see Appendix C).

| Method | SEVIR | | | Turbulence | | |
| | CRPS ↓ | | CSI ↑ | CRPS ↓ | | CSI ↑ |
| | $(w8, avg)$ | $(w8, max)$ | $(w5)$ | $(w8, avg)$ | $(w8, max)$ | $(w5)$ |
|---|---|---|---|---|---|---|
| DPM | 8.67 | 15.41 | 0.285 | 0.0313 | 0.0364 | 0.8960 |
| DyDiff (ours) | **7.62** | **13.56** | **0.319** | **0.0275** | **0.0315** | **0.8998** |

**Results.** Table 1 presents the numerical results. On both datasets, DyDiff consistently outperforms the standard DPM, achieving over a 12% reduction in CRPS on the Turbulence dataset. Further, figures 2 and 3 illustrate qualitative analyses. It is evident that DyDiff outputs more accurate predictions than standard DPM, particularly over longer time horizons.

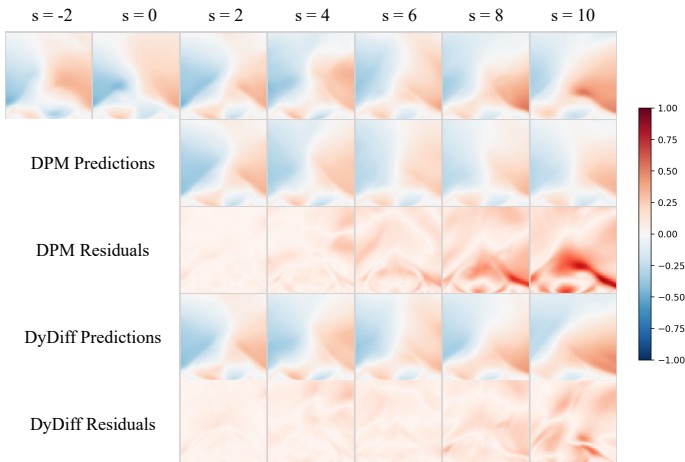

Figure 2: Visualization of predicted velocity fields on the Turbulence dataset. The top row displays the ground truth values. Residuals highlight the discrepancies between predictions and ground truths. Standard DPM predictions, characterized by two distinct positive regions (colored in red), do not align with physical laws. In contrast, Dynamical Diffusion yields more accurate predictions.

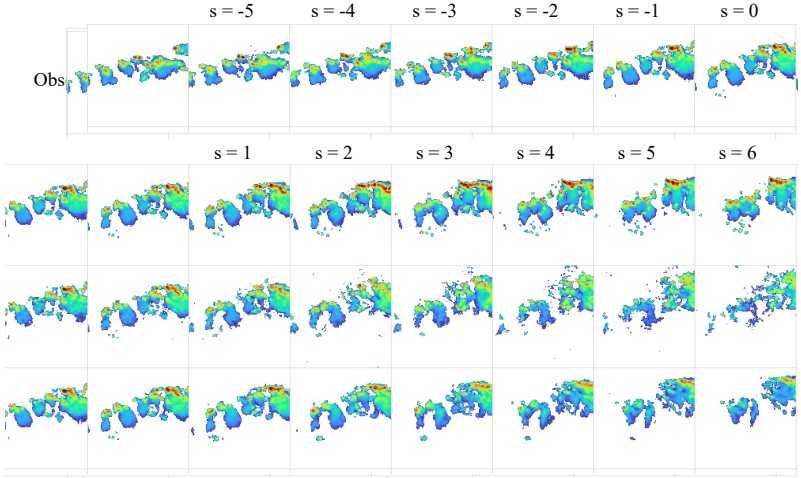

Figure 3: Visualization of predictions on the SEVIR dataset. The first row displays observational states, while the second row shows the corresponding ground truth. For longer prediction times, such as $s = 4$, standard diffusion models struggle to capture heavy-precipitation regions, particularly noticeable in the top right corner. In contrast, Dynamical Diffusion consistently provides more accurate predictions for these critical areas.

## 4.2 VIDEO PREDICTION

**Setup.** Next, we evaluate the performance of different methods on the BAIR (Ebert et al., 2017) and RoboNet (Dasari et al., 2019) datasets, which serve as benchmarks for assessing the model's ability to predict object movements in real-world scenarios. The BAIR robot pushing dataset consists of 43k training videos and 256 test videos. Each video records the motion of a robot as it pushes objects in a tabletop setting. The goal is to predict 15 future frames based on a single initial frame. The RoboNet dataset consists of 162k videos captured across 7 different robotic arms interacting with hundreds of objects in diverse environments and viewpoints. Following previous work (Yu et al., 2023), we use 256 videos for testing and predict 10 future frames based on 2 input frames. For both datasets, each frame has a resolution of $64 \times 64$ pixels. We report performance using four commonly adopted metrics: FVD (Unterthiner et al., 2018), PSNR (Huynh-Thu & Ghanbari, 2008), SSIM (Wang et al., 2004), and LPIPS (Zhang et al., 2018). Among these metrics, FVD

measures video-level consistency, while the other three are computed per image and reflect the average prediction accuracy.

Table 2: Video prediction results on the BAIR robot pushing and RoboNet dataset. LPIPS and SSIM scores are scaled by 100 for convenient display.

| BAIR | FVD↓ | PSNR↑ | SSIM↑ | LPIPS↓ | RoboNet | FVD↓ | PSNR↑ | SSIM↑ | LPIPS↓ |
|---|---|---|---|---|---|---|---|---|---|
| *action-free & 64×64 resolution* | | | | | *action-free & 64×64 resolution* | | | | |
| DPM | 72.0 | **21.0** | 83.8 | 9.2 | DPM | 92.9 | 24.9 | 83.9 | 8.2 |
| DyDiff (ours) | **67.4** | **21.0** | **84.0** | **9.0** | DyDiff (ours) | **81.7** | **25.1** | **84.2** | **7.9** |
| *action-conditioned & 64×64 resolution* | | | | | *action-conditioned & 64×64 resolution* | | | | |
| DPM | 48.5 | 25.9 | 92.0 | 4.5 | DPM | 77.0 | 26.4 | 87.3 | 6.0 |
| DyDiff (ours) | **45.0** | **26.2** | **92.4** | **4.2** | DyDiff (ours) | **67.7** | **26.5** | **87.5** | **5.9** |

**Results.** We present the experimental results in Table 2, covering both action-free and action-conditioned scenarios. Dynamical Diffusion consistently surpasses the standard DPM in all evaluated metrics, showing greater improvements in FVD, verifying its ability to make temporally consistent predictions. Qualitative outputs in Figure 4 illustrates that Dynamical Diffusion effectively addresses the artifact issues present in DPM for both background and foreground objects.

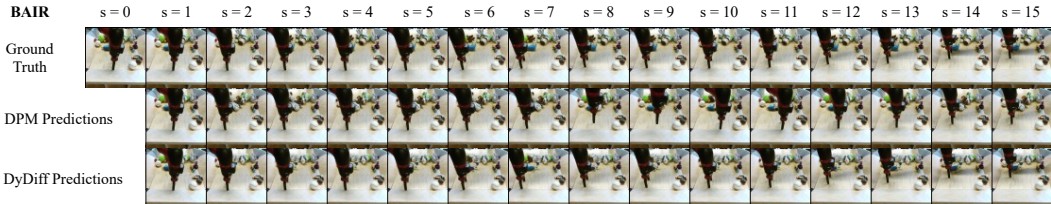

Figure 4: Visualization of action-conditioned predictions the BAIR dataset. Zoom in for details. The positions of robot arms under Dynamical Diffusion are more precise than standard DPM.

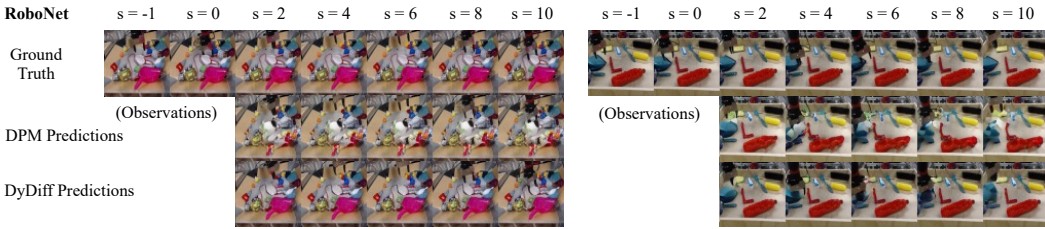

Figure 5: Visualization of action-conditioned predictions the RoboNet dataset. Zoom in for details. For standard diffusion models, *(left)* the pink shovel is missing, and *(right)* the red bottle is distorted. This indicates the potential temporal inconsistency of standard diffusion models. On the contrary, Dynamical Diffusion can generate consistent frames, especially for the background.

## 4.3 TIME SERIES FORECASTING

**Setup.** We further evaluate the model on six multivariate time series datasets: Exchange, Solar, Electricity, Traffic, Taxi, and Wikipedia. These datasets encompass time series with varying dimensionalities, domains, and sampling frequencies. We benchmark Dynamical Diffusion against the diffusion-based method TimeGrad (Rasul et al., 2021a), using TimeGrad's backbone and experimental setup. For evaluation, we employ the Summed CRPS (Matheson & Winkler, 1976).

**Results.** We present the experimental results in Table 3. Dynamical Diffusion significantly outperforms standard diffusion models in terms of CRPS$_{sum}$ on four out of six datasets (Solar, Traffic, Taxi, Wikipedia). For the remaining two datasets (Exchange, Electricity), the performance aligns

Table 3: Time series forecasting results on six benchmark datasets. CRPS$_{sum}$ is measured for its mean and standard deviation across five runs trained with different seeds.

| Method | CRPS$_{sum}$ ↓ | | | | | |
| --- | --- | --- | --- | --- | --- | --- |
| | Exchange | Solar | Electricity | Traffic | Taxi | Wikipedia |
| w/ DPM | $\mathbf{0.007}_{\pm 0.000}$ | $0.372_{\pm 0.064}$ | $\mathbf{0.021}_{\pm 0.002}$ | $0.042_{\pm 0.003}$ | $0.122_{\pm 0.012}$ | $0.070_{\pm 0.007}$ |
| w/ DyDiff | $\mathbf{0.007}_{\pm 0.000}$ | $\mathbf{0.316}_{\pm 0.010}$ | $0.023_{\pm 0.001}$ | $\mathbf{0.040}_{\pm 0.002}$ | $\mathbf{0.120}_{\pm 0.006}$ | $\mathbf{0.066}_{\pm 0.015}$ |

with the variance range of the baseline models. Overall, these results highlight the effectiveness of Dynamical Diffusion as a versatile predictive model across diverse datasets.

## 4.4 ANALYSIS

**Analysis on latents.** Dynamical Diffusion introduces novel forward and reverse processes, which affect the latents during inference stages. In Figure 6, we visualize and compare the latents of Dynamical Diffusion and the standard DPM. We also calculate the error between latents and final frames and plot the curve in Figure 7a. It is observed that DyDiff generates less noisy samples in an earlier denosing steps compared with standard diffusion model, especially for larger $s$.

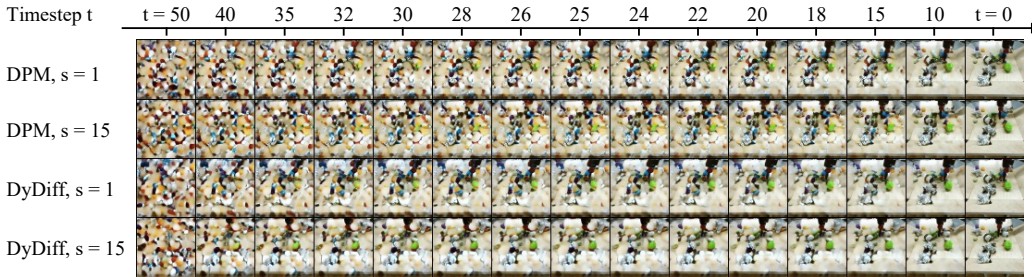

Figure 6: Visualization of latents during the inference process of the BAIR dataset, with timestep $t$ divided by 20. At the same timestep (such as $t = 22$), the backgrounds of frames generated by Dynamical Diffusion are consistent with the final predictions, while standard diffusion models hold noisier latents. Similar comparisons (such as $t = 28$) on Dynamical Diffusion show that frames with $s = 15$ are less noisy than $s = 1$ at a single timestep.

**Effect of dependent noises.** When using Dynamical Diffusion to predict multiple steps simultaneously, the forward process use non-independent noises $\widetilde{\epsilon}_t^s = \sqrt{\bar{\gamma}_t}\epsilon_t^s + \sqrt{1 - \bar{\gamma}_t}\widetilde{\epsilon}_{t-1}^s$, as illustrated in Theorem 3 and Algorithm 1. To further explore its necessity, we design an ablation study that uses independent noises $\epsilon_t^s$ instead of $\widetilde{\epsilon}_t^s$. Results are shown in Figure 7b. It is observed that when using independent noises $\epsilon_t^s$, the performance gets worse and even underperforms the baseline. Therefore, using non-independent noises is necessary for Dynamical Diffusion.

**Different gamma schedules.** For simplicity, Dynamical Diffusion adopt $\eta = 0.5$ as the default setting for the gamma schedule $\bar{\gamma}_t = \eta\bar{\alpha}_t + (1 - \eta)$. To further explore the sensitivity to hyperparameters, we conduct experiments using various $\eta$ with values in the set $\{0, 0.1, 0.5, 0.9, 1\}$, and report the results on the Turbulence dataset. Notably, $\eta = 0$ corresponds to the baseline of standard diffusion. Results are shown in Figure 7c. It is observed that $\eta \in \{0.1, 0.5, 0.9\}$ demonstrate similar performance and all significantly surpass the standard diffusion model, indicating the robustness of hyperparameter design in Dynamical Diffusion. Yet at $\eta = 1.0$, the model performance significantly drops and even underperforms the baseline, indicating that a schedule with $\bar{\gamma}_T \approx 0$ may not be effective, as discussed in Section 3.1.

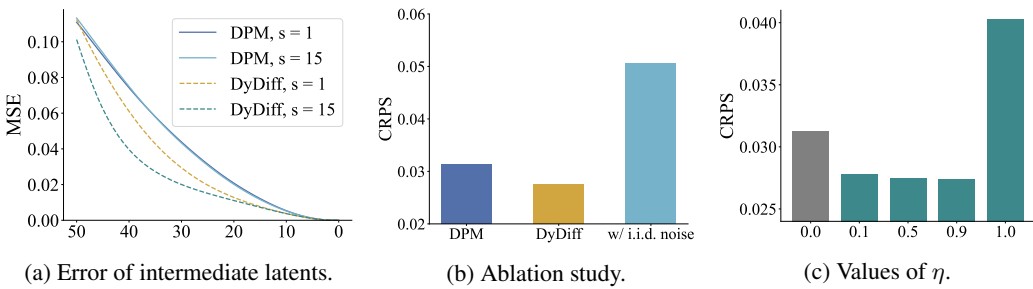

(a) Error of intermediate latents. (b) Ablation study. (c) Values of $\eta$.

Figure 7: Analysis experiments of Dynamical Diffusion.

## 5 RELATED WORK

**Studies on diffusion framework.** This paper studies around the diffusion equations, a topic extensively explored in the literature. Previous research has primarily concentrated on noise schedules (Nichol & Dhariwal, 2021; Karras et al., 2022; Liu et al., 2023), training objectives (Salimans & Ho, 2021; Karras et al., 2022), and efficient sampling techniques (Song et al., 2021; Lu et al., 2022). These methods aim to enhance the modeling of general data distributions including non-temporal modalities such as images. In contrast, our work presents a novel approach that explicitly incorporates temporal dynamics by modifying the diffusion equations.

**Deep predictive learning methods.** Predictive learning aims to forecast a system's future behavior by learning the underlying dynamics that drive its evolution. One common approach is to train the model to predict one step at a time and then unroll it autoregressively to make multi-step predictions. However, this method can be prone to compounding errors as the forecast horizon increases (de Bezenac et al., 2018; Scher & Messori, 2019; Chattopadhyay et al., 2020; Keisler, 2022; Bi et al., 2023). To address this, existing approaches focus on improving model architectures (Yan et al., 2021; Wang et al., 2022; Lam et al., 2022; Yu et al., 2023), unrolling models during training (Brandstetter et al., 2022; Pathak et al., 2022; HAN et al., 2022; Bi et al., 2023), or incorporating domain-specific knowledge (de Bezenac et al., 2018; Kochkov et al., 2021; Mamakoukas et al., 2023). Despite these efforts, performance in long-horizon prediction remains limited (Pathak et al., 2022). Alternatively, some methods forecast multiple steps simultaneously (Weyn et al., 2019; Brandstetter et al., 2022; Ravuri et al., 2021; Zhang et al., 2023), showing advantages over autoregressive methods in several contexts (Voleti et al., 2022; Gao et al., 2023). In both lines of work, generative models, such as generative adversarial networks (GANs) and diffusion models, have been leveraged for their superior ability to model distributions, enhancing prediction quality (Rasul et al., 2021a; Zhang et al., 2023; Mardani et al., 2023; Gao et al., 2023; Pathak et al., 2024). Our work specifically focuses on generating multi-step predictions simultaneously with diffusion models, a topic that has gained increasing attention in the research community.

**Predictive learning with diffusion models.** To enhance predictive learning with diffusion models, Ho et al. (2022); Blattmann et al. (2023a); Voleti et al. (2022); Gao et al. (2023); Rasul et al. (2021a) design specific predictive model architectures for different modalities. Wu et al. (2023); Ruhe et al. (2024); Chen et al. (2024a) propose state-wise timestep schedules. Notably, in these methods, both the forward and reverse processes remain consistent with standard formulations. Therefore, our work serves as a complement to existing approaches.

## 6 CONCLUSION

In this paper, we investigate temporal predictive learning using diffusion models and highlight the underexplored challenge of integrating temporal dynamics into the diffusion process. For this purpose, we introduce Dynamical Diffusion, a theoretically guaranteed framework that explicitly models temporal transitions at each diffusion step. Dynamical Diffusion introduces a simple yet efficient design that adds noises to the combination of the current state and historical states, which is further learned by a denoising process. Experiments on various tasks, including scientific spatiotemporal forecasting, video prediction, and time series forecasting, demonstrate that Dynamical Diffusion consistently enhances performance in general predictive learning.

## ACKNOWLEDGEMENT

This work was supported by the National Natural Science Foundation of China (U2342217 and 62021002), the BNRist Project, and the National Engineering Research Center for Big Data Software. We also gratefully acknowledge the contributions of Yuchen Zhang for helpful discussion and Jincheng Zhong for the Overleaf premium plan.

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

## A MATHEMATICAL PROOF

### A.1 DERIVATION OF FORWARD PROCESS

In this subsection we give the proof of the form of $\mathbf{x}_s^t$ in Equation (4).

*Proof.* By mathematical induction on $s$. With definitions in Equation (3)

$$\mathbf{x}_t^s = \sqrt{\bar{\gamma}_t} \cdot \left( \sqrt{\bar{\alpha}_t} \mathbf{x}_0^s + \sqrt{1 - \bar{\alpha}_t} \boldsymbol{\epsilon}_t^s \right) + \sqrt{1 - \bar{\gamma}_t} \cdot \mathbf{x}_t^{s-1},$$

Equation (5)

$$\texttt{Dynamics}\left(\mathbf{x}_0^{-P:s}, \bar{\gamma}_t\right) = \sqrt{\bar{\gamma}_t} \cdot \mathbf{x}_0^s + \sqrt{1 - \bar{\gamma}_t} \cdot \texttt{Dynamics}\left(\mathbf{x}_0^{-P:s-1}, \bar{\gamma}_t\right),$$

and the boundary condition as the start state,

$$\mathbf{x}_t^{-P} = \sqrt{\bar{\alpha}_t} \mathbf{x}_0^{-P} + \sqrt{1 - \bar{\alpha}_t} \boldsymbol{\epsilon}_t^{-P},$$

it holds

$$\begin{aligned}
\mathbf{x}_t^s &= \sqrt{\bar{\gamma}_t} \cdot \left( \sqrt{\bar{\alpha}_t} \mathbf{x}_0^s + \sqrt{1 - \bar{\alpha}_t} \boldsymbol{\epsilon}_t^s \right) + \sqrt{1 - \bar{\gamma}_t} \cdot \mathbf{x}_t^{s-1} \\
&= \sqrt{\bar{\gamma}_t} \cdot \left( \sqrt{\bar{\alpha}_t} \mathbf{x}_0^s + \sqrt{1 - \bar{\alpha}_t} \boldsymbol{\epsilon}_t^s \right) \\
&\quad + \sqrt{1 - \bar{\gamma}_t} \cdot \left( \sqrt{\bar{\alpha}_t} \cdot \texttt{Dynamics}\left(\mathbf{x}_0^{-P:s-1}; \bar{\gamma}_t\right) + \sqrt{1 - \bar{\alpha}_t} \cdot \widetilde{\boldsymbol{\epsilon}}_t^{s-1} \right) \\
&= \sqrt{\bar{\alpha}_t} \cdot \left( \sqrt{\bar{\gamma}_t} \cdot \mathbf{x}_0^s + \sqrt{1 - \bar{\gamma}_t} \cdot \texttt{Dynamics}\left(\mathbf{x}_0^{-P:s-1}; \bar{\gamma}_t\right) \right) \\
&\quad + \sqrt{1 - \bar{\alpha}_t} \cdot \left( \sqrt{\bar{\gamma}_t} \cdot \boldsymbol{\epsilon}_t^s + \sqrt{1 - \bar{\gamma}_t} \cdot \widetilde{\boldsymbol{\epsilon}}_t^{s-1} \right) \\
&= \sqrt{\bar{\alpha}_t} \cdot \texttt{Dynamics}\left(\mathbf{x}_0^{-P:s}; \bar{\gamma}_t\right) + \sqrt{1 - \bar{\alpha}_t} \cdot \widetilde{\boldsymbol{\epsilon}}_t^s.
\end{aligned}$$

Since $\widetilde{\boldsymbol{\epsilon}}_t^{s-1}$ and $\boldsymbol{\epsilon}_t^s$ are independent normal noise, it satisfies

$$\widetilde{\boldsymbol{\epsilon}}_t^s = \sqrt{\bar{\gamma}_t} \boldsymbol{\epsilon}_t^s + \sqrt{1 - \bar{\gamma}_t} \widetilde{\boldsymbol{\epsilon}}_t^{s-1} \sim \mathcal{N}(\mathbf{0}, \mathbf{I}),$$

which completes the proof. $\qquad\square$

### A.2 CASE WHEN $S = 1$

**DDIM-like sampler.** We follow the proof in DDIM (Song et al., 2021)

*Proof.* Define the following non-Markovian forward process:

$$q(\mathbf{x}_{1:T}^1 | \mathbf{x}_0^{-P:1}) = q(\mathbf{x}_T^1 | \mathbf{x}_0^{-P:1}) \prod_{t=2}^{T} q(\mathbf{x}_{t-1}^1 | \mathbf{x}_t^1, \mathbf{x}_0^{-P:1})$$

with $q(\mathbf{x}_T^1 | \mathbf{x}_0^{-P:1}) = \mathcal{N}\left( \sqrt{\bar{\alpha}_T} \cdot \texttt{Dynamics}(\mathbf{x}_0^{-P:1}, \bar{\gamma}_T), (1 - \bar{\alpha}_T)\mathbf{I} \right),$

$$q(\mathbf{x}_{t-1}^1 | \mathbf{x}_t^1, \mathbf{x}_0^{-P:1}) = \mathcal{N}\Bigg( \sqrt{\bar{\alpha}_{t-1}} \cdot \texttt{Dynamics}(\mathbf{x}_0^{-P:1}, \bar{\gamma}_{t-1}) +$$

$$\sqrt{1 - \bar{\alpha}_{t-1} - \sigma_t^2} \cdot \frac{\mathbf{x_t} - \sqrt{\bar{\alpha}_t} \cdot \texttt{Dynamics}(\mathbf{x}_0^{-P:1}, \bar{\gamma}_t)}{\sqrt{1 - \bar{\alpha}_t}}, \sigma_t^2 \mathbf{I} \Bigg),$$

then it suffies to prove $q(\mathbf{x}_t^1 | \mathbf{x}_0^{-P:0}) = \mathcal{N}\left( \sqrt{\bar{\alpha}_t} \cdot \texttt{Dynamics}(\mathbf{x}_0^{-P:1}, \bar{\gamma}_t), (1 - \bar{\alpha}_t)\mathbf{I} \right)$. By mathematical induction on $t$ from $T - 1$ to $1$, we have

$$q(\mathbf{x}_t^1 | \mathbf{x}_0^{-P:1}) = \mathcal{N}\left( \sqrt{\bar{\alpha}_t} \cdot \texttt{Dynamics}(\mathbf{x}_0^{-P:1}, \bar{\gamma}_t), (1 - \bar{\alpha}_t)\mathbf{I} \right),$$

$$q(\mathbf{x}_{t-1}^1 | \mathbf{x}_t^1, \mathbf{x}_0^{-P:1}) = \mathcal{N}\Bigg( \sqrt{\bar{\alpha}_{t-1}} \cdot \texttt{Dynamics}(\mathbf{x}_0^{-P:1}, \bar{\gamma}_{t-1}) +$$

$$\sqrt{1 - \bar{\alpha}_{t-1} - \sigma_t^2} \cdot \frac{\mathbf{x_t} - \sqrt{\bar{\alpha}_t} \cdot \texttt{Dynamics}(\mathbf{x}_0^{-P:1}, \bar{\gamma}_t)}{\sqrt{1 - \bar{\alpha}_t}}, \sigma_t^2 \mathbf{I} \Bigg),$$

then by letting $a \leftarrow \mathbf{x}_t^1, b \leftarrow \mathbf{x}_{t-1}^1$, and $c \leftarrow \mathbf{x}_0^{-P:1}$ as a global condition, according to Eq. (2.115) (Bishop, 2006),

$$q(\mathbf{x}_{t-1}^1|\mathbf{x}_0^{-P:1}) = \int_{\mathbf{x}_t^1} q(\mathbf{x}_t^1|\mathbf{x}_0^{-P:1})q(\mathbf{x}_{t-1}^1|\mathbf{x}_t^1, \mathbf{x}_0^{-P:1})d\mathbf{x}_t^1$$

$$q_c(b) = \int_a q_c(a)q_c(b|a)da$$

is also a Gaussian with

$$\boldsymbol{\mu}_{t-1} = \sqrt{\bar{\alpha}_{t-1}} \cdot \texttt{Dynamics}(\mathbf{x}_0^{-P:1}, \bar{\gamma}_{t-1})$$
$$+ \sqrt{1 - \bar{\alpha}_{t-1} - \sigma_t^2} \cdot \frac{\sqrt{\bar{\alpha}_t} \cdot \texttt{Dynamics}(\mathbf{x}_0^{-P:1}, \bar{\gamma}_t) - \sqrt{\bar{\alpha}_t} \cdot \texttt{Dynamics}(\mathbf{x}_0^{-P:1}, \bar{\gamma}_t)}{\sqrt{1 - \bar{\alpha}_t}}$$
$$= \sqrt{\bar{\alpha}_{t-1}} \cdot \texttt{Dynamics}(\mathbf{x}_0^{-P:1}, \bar{\gamma}_{t-1}),$$
$$\boldsymbol{\Sigma}_{t-1} = \sigma_t^2 \mathbf{I} + \frac{1 - \bar{\alpha}_{t-1} - \sigma_t^2}{1 - \bar{\alpha}_t}(1 - \bar{\alpha}_t)\mathbf{I} = (1 - \bar{\alpha}_{t-1})\mathbf{I},$$

which completes the proof. $\qquad\square$

**DDPM-like sampler.**    We follow the proof in DDPM (Ho et al., 2020).

*Proof.* The proof is structured in two steps. First, define the following Markovian forward process:

$$q\left(\mathbf{x}_t^1|\mathbf{x}_{t-1}^1, \mathbf{x}_0^{-P:1}\right) \sim \mathcal{N}(\boldsymbol{\mu}_t, \boldsymbol{\Sigma}_t)$$

$$\boldsymbol{\mu}_t = \sqrt{\alpha_t \gamma_t} \cdot \mathbf{x}_{t-1} + \sqrt{\bar{\alpha}_t} \left( \sqrt{1 - \bar{\gamma}_t} \cdot \texttt{Dynamics}(\mathbf{x}_0^{-P:0}; \bar{\gamma}_t) - \sqrt{\gamma_t - \bar{\gamma}_t} \cdot \texttt{Dynamics}(\mathbf{x}_0^{-P:0}; \bar{\gamma}_{t-1}) \right)$$

$$\boldsymbol{\Sigma}_t = (1 - \alpha_t \gamma_t - \bar{\alpha}_t(1 - \gamma_t)) \mathbf{I}$$

where $\bar{\alpha}_t = \prod_{i=1}^t \alpha_i, \bar{\gamma}_t = \prod_{i=1}^t \gamma_i$. It suffies to prove the marginal distribution

$$q(\mathbf{x}_t^1|\mathbf{x}_0^{-P:1}) = \mathcal{N}\left(\sqrt{\bar{\alpha}_t} \cdot \texttt{Dynamics}(\mathbf{x}_0^{-P:1}, \bar{\gamma}_t), (1 - \bar{\alpha}_t)\mathbf{I}\right).$$

By mathematical induction on $t$ from 1 to $T - 1$, we have $q(\mathbf{x}_{t+1}^1|\mathbf{x}_t^1, \mathbf{x}_0^{-P:0})$ and $q(\mathbf{x}_t^1|\mathbf{x}_0^{-P:1})$ are Gaussian distributions, and thus by letting $a \leftarrow \mathbf{x}_t^1, b \leftarrow \mathbf{x}_{t+1}^1$, and $c \leftarrow \mathbf{x}_0^{-P:1}$ as a global condition, according to Eq. (2.115) (Bishop, 2006),

$$q(\mathbf{x}_{t+1}^1|\mathbf{x}_0^{-P:1}) = \int_{\mathbf{x}_t^1} q(\mathbf{x}_t^1|\mathbf{x}_0^{-P:1})q(\mathbf{x}_{t+1}^1|\mathbf{x}_t^1, \mathbf{x}_0^{-P:1})d\mathbf{x}_t^1$$

$$q_c(b) = \int_a q_c(a)q_c(b|a)da$$

is also a Gaussian distribution with

$$\boldsymbol{\mu}_{t+1} = \sqrt{\bar{\alpha}_{t+1}\gamma_t} \cdot \texttt{Dynamics}(\mathbf{x}_0^{-P:1}, \bar{\gamma}_t)$$
$$+ \sqrt{\bar{\alpha}_{t+1}} \left( \sqrt{1 - \bar{\gamma}_{t+1}} \cdot \texttt{Dynamics}(\mathbf{x}_0^{-P:0}; \bar{\gamma}_{t+1}) - \sqrt{\gamma_{t+1} - \bar{\gamma}_{t+1}} \cdot \texttt{Dynamics}(\mathbf{x}_0^{-P:0}; \bar{\gamma}_t) \right)$$
$$= \sqrt{\bar{\alpha}_t} \cdot \texttt{Dynamics}(\mathbf{x}_0^{-P:1}, \bar{\gamma}_t)$$
$$\boldsymbol{\Sigma}_{t+1} = (1 - \alpha_{t+1}\gamma_{t+1} - \bar{\alpha}_{t+1}(1 - \gamma_{t+1}))\mathbf{I} + \alpha_{t+1}\gamma_{t+1}(1 - \bar{\alpha}_t)\mathbf{I} = (1 - \bar{\alpha}_{t+1})\mathbf{I}.$$

Next, we consider the posterior distribution for the reverse process. Since $q(\mathbf{x}_t^1|\mathbf{x}_{t-1}^1, \mathbf{x}_0^{-P:1})$ and $q(\mathbf{x}_{t-1}^1|\mathbf{x}_0^{-P:1})$ are both Gaussians, by letting $a \leftarrow \mathbf{x}_{t-1}^1, b \leftarrow \mathbf{x}_t^1$, and $c \leftarrow \mathbf{x}_0^{-P:1}$ as a global condition, according to Eq. (2.116) (Bishop, 2006),

$$q(\mathbf{x}_{t-1}^1|\mathbf{x}_t^1, \mathbf{x}_0^{-P:1}) = \frac{q(\mathbf{x}_t^1|\mathbf{x}_{t-1}^1, \mathbf{x}_0^{-P:0})q(\mathbf{x}_t^1|\mathbf{x}_0^{-P:1})}{q(\mathbf{x}_{t-1}^1|\mathbf{x}_0^{-P:1})}$$

$$q_c(a|b) = \frac{q_c(b|a)q_c(a)}{q_c(b)}$$

is also a Gaussian distribution. Therefore, the existence of the reverse process is proved, where the mean and variance could be derived accordingly. $\qquad\square$

### A.3 Case when $S > 1$

*Proof.* For $S > 1$, by definition in Equation (3), let

$$\mathbf{y}_t^1 = \mathbf{x}_t^1,$$

$$\mathbf{y}_t^s = \frac{\mathbf{x}_t^s - \sqrt{1 - \bar{\gamma}_t} \cdot \mathbf{x}_t^{s-1}}{\sqrt{\bar{\gamma}_t}}, \qquad \forall 1 < s \leq S,$$

then $\mathbf{y}_t^s = \sqrt{\bar{\alpha}_t}\mathbf{x}_0^s + \sqrt{1 - \bar{\alpha}_t}\boldsymbol{\epsilon}_t^s$, satisfying the same Guassian distribution as standard diffusion models and independent with $\mathbf{y}_t^{s'}, \forall s' \neq s$. Thus

$$q(\mathbf{y}_t^{1:S}|\mathbf{x}_0^{-P:S}) = q(\mathbf{x}_t^1|\mathbf{x}_0^{-P:1}) \prod_{s=2}^{S} q(\mathbf{y}_t^s|\mathbf{x}_0^s).$$

By defining the distribution of $\mathbf{x}_t^1$ as Appendix A.2 and $\mathbf{y}_t^s, 1 < s \leq S$ as DDPM/DDIM for standard diffusion models, the reverse process gets proved by combining these independent latents together. $\square$

*Remark.* Following the proof, seemingly there is no need to add noise on dynamics for $s > 1$ in theoretical view. In practice, the manner that remains dynamics could potentially help model generalization.

## B Algorithm for Inverse Dynamics

In this section we discuss the calculation of the inverse dynamics used in the inference process, i.e., calculate $\mathbf{x}_0^s$ when given $\texttt{Dynamics}(\mathbf{x}_0^{-P:k}; \bar{\gamma})$ for $-P \leq k \leq s$. From Equation (5), we have

$$\mathbf{x}_0^s = \frac{\texttt{Dynamics}(\mathbf{x}_0^{-P:s}; \bar{\gamma}) - \sqrt{1 - \bar{\gamma}} \cdot \texttt{Dynamics}(\mathbf{x}_0^{-P:s-1}; \bar{\gamma})}{\sqrt{\bar{\gamma}}}.$$

The pseudocode of calculating inverse dynamics is shown in Algorithm 3.

---

**Algorithm 3** Inverse Dynamics

---

1: **procedure** InverseDynamics($\mathbf{x}_{\text{dyn}}^{L:R}, \bar{\gamma}$)
2:      $\mathbf{x}_0^L \leftarrow \mathbf{x}_{\text{dyn}}^L$
3:      **for** $s$ in $[L+1, R]$ **do**
4:          $\mathbf{x}_0^s \leftarrow \left(\mathbf{x}_{\text{dyn}}^s - \sqrt{1 - \bar{\gamma}}\mathbf{x}_{\text{dyn}}^{s-1}\right) / \sqrt{\bar{\gamma}}$
5:      **end for**
6:      **return** $\mathbf{x}_0^{L:R}$
7: **end procedure**

---

## C Implementation Details

### C.1 Spatiotemporal forecasting and video prediction

**Training details.** For the benchmark datasets, including BAIR, RoboNet, Turbulence, and SE-VIR, we utilize the state-of-the-art architecture of Stable Video Diffusion (Blattmann et al., 2023a). Specifically, we first train frame-wise VAEs from scratch. In line with Blattmann et al., we also incorporate an adversarial discriminator during VAE training to enhance reconstruction quality. The spatial downsampling ratio for the VAE is set to $4 \times 4$ across all datasets. Once trained, the VAE encodes the original data into a latent space with a channel size of 3, and all diffusion processes are carried out in this latent space. We adopt a 3D UNet as the diffusion model. All diffusion models are also trained from scratch. Table 4 presents the detailed hyperparameters on these datasets.

Table 4: Hyperparameters of DyDiff training.

| DyDiff | Low-resolution $(64 \times 64)$ | | | High-resolution $(128 \times 128)$ |
| --- | --- | --- | --- | --- |
| | BAIR | RoboNet | Turbulence | SEVIR |
| Input channel | 3 | 3 | 2 | 1 |
| Prediction length | 15 | 10 | 11 | 6 |
| Observation length | 1 | 2 | 4 | 7 |
| Training steps | $5 \times 10^5$ | $5 \times 10^5$ | $3 \times 10^5$ | $4 \times 10^5$ |
| VAE channels | $[128, 256, 512]$ | | | |
| VAE downsampling ratio | $4 \times 4$ | | | |
| VAE kl weighting | $1 \times 10^{-6}$ | | | |
| Latent channel | 3 | | | |
| SVD channels | $[64, 128, 256, 256]$ | | | |
| Batch size | 16 | | | |
| Learning rate | $1 \times 10^{-4}$ | | | |
| LR Schedule | Constant | | | |
| Optimizer | Adam | | | |

**Evaluation metrics.** We evaluate each method using commonly employed metrics, as outlined below:

- **Critical Success Index (CSI)** (Schaefer, 1990) quantifies the accuracy of binary predictive decisions. Following (Chen et al., 2024b), we apply a spatial window around each grid for neighborhood-based evaluation (Jolliffe & Stephenson, 2012) to evaluate the "closeness" of the forecasts. The window size ($w$) is set to 5, and average pooling ($avg$) is used within the window.

- **Continuous Ranked Probability Score (CRPS)** (Gneiting & Raftery, 2007) measures the alignment between probabilistic forecasts and ground truth data. To compute CRPS, the model generates multiple forecasts, allowing the score to capture the entire probability distribution. Following (Chen et al., 2024b), we calculate the neighborhood-based CRPS using a window size of 8, and report results for two pooling modes: average pooling ($avg$) and max pooling ($max$).

- **The Peak Signal-to-Noise Ratio (PSNR)** (Huynh-Thu & Ghanbari, 2008) measures the ratio between the maximum possible signal power (in this case, an image or video) and the power of the noise or distortion affecting it. A higher PSNR indicates less distortion and a closer match to the original image.

- **The Structural Similarity Index Measure (SSIM)** (Wang et al., 2004) is a widely used metric for evaluating the quality of images and video frames by assessing their perceived structural similarity. The SSIM score ranges from -1 to 1, with 1 indicating perfect structural similarity and lower values indicating greater dissimilarity. To better present the results, we scale the SSIM score by a factor of 100.

- **The Learned Perceptual Image Patch Similarity (LPIPS)** (Zhang et al., 2018) evaluates the similarity between two images by passing them through a pretrained neural network. The network extracts features from both images, and the LPIPS score is calculated based on the distance between these feature representations. A smaller LPIPS score indicates higher similarity. Similar to SSIM, we also scale the LPIPS score by 100.

- **The Fréchet Video Distance (FVD)** (Unterthiner et al., 2018) is based on the Fréchet distance, a mathematical measure that computes the distance between two distributions. For FVD, these distributions represent the feature space of real and generated videos extracted by a neural network. Unlike the metrics mentioned above, FVD incorporates the temporal dimension of videos, making it more suitable for evaluating video generation models.

**Sampling protocals.** During inference, we use DDIM sampler with 50 steps for both the standard DPM and our proposed Dynamical Diffusion. For video prediction benchmarks, including

BAIR and RoboNet, following prior works (Gupta et al., 2023; Wu et al., 2024), we account for the stochastic nature of video prediction by sampling 100 future trajectories per test video and selecting the best one for the final PSNR, SSIM, and LPIPS scores. For FVD, we use all 100 samples. For scientific spatiotemporal forecasting tasks, including Turbulence and SEVIR, we generate 8 predictions for each test sample to compute CRPS and CSI, in line with prior work (Chen et al., 2024b).

### C.2 TIME SERIES FORECASTING

**Training details.** For time series forecasting tasks, we follow the benchmark of TimeGrad (Rasul et al., 2021a), which is a framework to apply diffusion models with the next-token prediction paradiam in time series forecasting. For implementation of Dynamical Diffusion, we set $P = 0$, i.e., apply dynamics on only the latest state to match the Markovian properties in the RNN used in TimeGrad. Since time series have greater volatility, we set $1 - \gamma_t = 0.3(1 - \alpha_t)$ for training and inference stability. We use exactly the same model architecture as TimeGrad. Since TimeGrad does not provide publically available reproducible setups, we carefully tune the baselines and Dynamical Diffusion for the best performance on each dataset. All datasets are available through GluonTS (Alexandrov et al., 2019), with detailed information shown in Table 5.

Table 5: Properties of time series forecasting datasets.

| Dataset | Dimension | Domain | Frequency | Steps | Prediction length |
|---|---|---|---|---|---|
| Exchange | 8 | $\mathbb{R}^+$ | BUSINESS DAY | 6,071 | 30 |
| Solar | 137 | $\mathbb{R}^+$ | HOUR | 7,009 | 24 |
| Electricity | 370 | $\mathbb{R}^+$ | HOUR | 5,833 | 24 |
| Traffic | 963 | (0,1) | HOUR | 4,001 | 24 |
| Taxi | 1,214 | $\mathbb{N}$ | 30-MIN | 1,488 | 24 |
| Wikipedia | 2,000 | $\mathbb{N}$ | DAY | 792 | 30 |

**Evaluation metrics.** Following TimeGrad (Rasul et al., 2021a), we employ the Summed CRPS (Matheson & Winkler, 1976) to capture the joint effect, where score is evaluated based on the sum of predicted distribution.

**Sampling protocals.** We use DDIM sampler with 50 steps for the standard DPM and Dynamical Diffusion. For calculating the Summed CRPS, we generate 100 predictions for each test sample.

## D MORE EXPERIMENT RESULTS

### D.1 SCIENTIFIC SPATIOTEMPORAL FORECASTING

In this section, we further experiment with Diffusion Transformers (DiT) (Peebles & Xie, 2023). Table 6 presents the results of the Turbulence Flow dataset. We follow the same evaluation protocols outlined in Appendix C for these experiments. The results demonstrate that DyDiff significantly outperforms standard diffusion models, confirming its general applicability.

Table 6: Scientific spatiotemporal forecasting results on the Turbulence Flow dataset.

| Backbone | Method | CRPS $\downarrow$ | | CSI $\uparrow$ |
|---|---|---|---|---|
| | | $(w8, avg)$ | $(w8, max)$ | $(w5)$ |
| SVD | DPM | 0.0313 | 0.0364 | 0.8960 |
| | DyDiff (ours) | **0.0275** | **0.0315** | **0.8998** |
| DiT | DPM | 0.0434 | 0.0480 | 0.8403 |
| | DyDiff (ours) | **0.0358** | **0.0395** | **0.8548** |

## D.2 VIDEO PREDICTION

In this section, we provide additonal comparisons with state-of-the-art deterministic models, including VideoGPT (Yan et al., 2021), MaskViT (Gupta et al., 2023), FitVid (Babaeizadeh et al., 2021), MAGVIT (Yu et al., 2023), SVG (Villegas et al., 2019), GHVAE (Wu et al., 2021), and iVideoGPT (Wu et al., 2024) on video prediction benchmarks. Table 7 and 8 present the results on BAIR and RoboNet datasets, respectively. On the BAIR dataset, DyDiff demonstrates comparable performance in the action-free scenario and significantly outperforms previous deterministic methods in the action-conditioned scenario. However, the RoboNet dataset, characterized by its diverse object motion trajectories, poses a substantial challenge for both DPM and DyDiff, with both methods falling short in performance. Notably, these methods employ networks with significantly more parameters than ours—for instance, iVideoGPT contains 114M parameters, and MaskViT contains 189M, compared to our model's 63M parameters. Besides, some methods involve an addition pretraining process (Wu et al., 2024).

Table 7: Addition comparison with deterministic methods on BAIR dataset. "-" marks that the value is not reported in the original papers. LPIPS and SSIM scores are scaled by 100 for convenient display.

| BAIR | FVD↓ | PSNR↑ | SSIM↑ | LPIPS↓ |
|---|---|---|---|---|
| *action-free & 64×64 resolution* | | | | |
| VideoGPT 2021 | 103.3 | - | - | - |
| MaskViT 2023 | 93.7 | - | - | - |
| FitVid 2021 | 93.6 | - | - | - |
| MCVD 2022 | 89.5 | 16.9 | 78.0 | - |
| MAGVIT 2023 | **62.0** | 19.3 | 78.7 | 12.3 |
| iVideoGPT 2024 | 75.0 | 20.4 | 82.3 | 9.5 |
| DPM | 72.0 | **21.0** | 83.8 | 9.2 |
| DyDiff (ours) | 67.4 | **21.0** | **84.0** | **9.0** |
| *action-conditioned & 64×64 resolution* | | | | |
| MaskViT 2023 | 70.5 | - | - | - |
| iVideoGPT 2024 | 60.8 | 24.5 | 90.2 | 5.0 |
| DPM | 48.5 | 25.9 | 92.0 | 4.5 |
| DyDiff (ours) | **45.0** | **26.2** | **92.4** | **4.2** |

Table 8: Addition comparison with deterministic methods on RoboNet dataset. LPIPS and SSIM scores are scaled by 100 for convenient display.

| RoboNet | FVD↓ | PSNR↑ | SSIM↑ | LPIPS↓ |
|---|---|---|---|---|
| *action-conditioned & 64×64 resolution* | | | | |
| MaskViT 2023 | 133.5 | 23.2 | 80.5 | 4.2 |
| SVG 2019 | 123.2 | 23.9 | 87.8 | 6.0 |
| GHVAE 2021 | 95.2 | 24.7 | 89.1 | 3.6 |
| FitVid 2021 | **62.5** | **28.2** | 89.3 | **2.4** |
| iVideoGPT 2024 | 63.2 | 27.8 | **90.6** | 4.9 |
| DPM | 92.9 | 24.9 | 83.9 | 8.2 |
| DyDiff (ours) | 81.7 | 25.1 | 84.2 | 7.9 |

## D.3 TIME SERIES FORECASTING

This section shows additonal comparisons with the standard diffusion model baseline (TimeGrad (Rasul et al., 2021a)) and the state-of-the-art time series forecasting models, including VES (Hyndman et al., 2008), VAR (Lütkepohl, 2005)(-Lasso), GARCH (van der Weide, 2002), DeepAR (Salinas et al., 2020), LSTP/GP-Copula (Salinas et al., 2019), KVAE (Krishnan et al., 2017), NKF (de Bézenac et al., 2020) and Transformer-MAF (Rasul et al., 2021b). As demonstrated in Table 9, diffusion-based forecasting models fundamentally achieve similar or better performance

compared with deterministic models. Furthermore, Dynamical Diffusion generally outperforms standard diffusion baselines.

Table 9: Addition comparison with deterministic methods on Time Series dataset. CRPS$_{sum}$ (lower indicates better) is measured for its mean and standard deviation across five runs trained with different seeds. "-" marks that the value is not reported in the original papers.

| Method | Exchange | Solar | Electricity | Traffic | Taxi | Wikipedia |
|---|---|---|---|---|---|---|
| VES 2008 | $\mathbf{0.005}_{\pm 0.000}$ | $0.900_{\pm 0.003}$ | $0.880_{\pm 0.004}$ | $0.350_{\pm 0.002}$ | - | - |
| VAR 2005 | $\mathbf{0.005}_{\pm 0.000}$ | $0.830_{\pm 0.006}$ | $0.039_{\pm 0.001}$ | $0.290_{\pm 0.001}$ | - | - |
| VAR-Lasso 2005 | $0.012_{\pm 0.000}$ | $0.510_{\pm 0.006}$ | $0.025_{\pm 0.000}$ | $0.150_{\pm 0.002}$ | - | $3.100_{\pm 0.004}$ |
| GARCH 2002 | $0.023_{\pm 0.000}$ | $0.880_{\pm 0.002}$ | $0.190_{\pm 0.001}$ | $0.370_{\pm 0.001}$ | - | - |
| DeepAR 2020 | - | $0.336_{\pm 0.014}$ | $0.023_{\pm 0.001}$ | $0.055_{\pm 0.003}$ | - | $0.127_{\pm 0.042}$ |
| LSTM-Copula 2019 | $\underline{0.007}_{\pm 0.000}$ | $0.319_{\pm 0.011}$ | $0.064_{\pm 0.008}$ | $0.103_{\pm 0.006}$ | $0.326_{\pm 0.007}$ | $0.241_{\pm 0.033}$ |
| GP-Copula 2019 | $\underline{0.007}_{\pm 0.000}$ | $0.337_{\pm 0.024}$ | $0.025_{\pm 0.002}$ | $0.078_{\pm 0.002}$ | $0.208_{\pm 0.183}$ | $0.086_{\pm 0.004}$ |
| KVAE 2017 | $0.014_{\pm 0.002}$ | $0.340_{\pm 0.025}$ | $0.051_{\pm 0.019}$ | $0.100_{\pm 0.005}$ | - | $0.095_{\pm 0.012}$ |
| NKF 2020 | - | $0.320_{\pm 0.020}$ | $\mathbf{0.016}_{\pm 0.001}$ | $0.100_{\pm 0.002}$ | - | - |
| Transformer-MAF 2021b | $\mathbf{0.005}_{\pm 0.003}$ | $\mathbf{0.301}_{\pm 0.014}$ | $\underline{0.021}_{\pm 0.000}$ | $0.056_{\pm 0.001}$ | $0.179_{\pm 0.002}$ | $\mathbf{0.063}_{\pm 0.003}$ |
| TimeGrad w/ DPM | $\underline{0.007}_{\pm 0.000}$ | $0.372_{\pm 0.064}$ | $\underline{0.021}_{\pm 0.002}$ | $\underline{0.042}_{\pm 0.003}$ | $\underline{0.122}_{\pm 0.012}$ | $0.070_{\pm 0.007}$ |
| TimeGrad w/ DyDiff | $\underline{0.007}_{\pm 0.000}$ | $\underline{0.316}_{\pm 0.010}$ | $0.023_{\pm 0.001}$ | $\mathbf{0.040}_{\pm 0.002}$ | $\mathbf{0.120}_{\pm 0.006}$ | $\underline{0.066}_{\pm 0.015}$ |

