# OpenReview forum: "Dynamical Diffusion: Learning Temporal Dynamics with Diffusion Models"
_ICLR.cc/2025/Conference — ICLR 2025 Poster_

### Official Review · Reviewer_hYkb · 2024-10-27

**Soundness:** 3
**Presentation:** 3
**Contribution:** 3
**Rating:** 6
**Confidence:** 3

**Summary:**

The authors propose a new diffusion model for time series data incoporating the temporal dependencies between states in the forward and backward processes. Extensive empirical evidence shows that the model indeed outperforms regular denoising diffusion most notably on the SEVIR and Turbulence Flow datasets.

**Strengths:**

- **A clear problem and a sound solution**: The paper identifies a significant gap in how current diffusion models handle temporal data - they treat predictive learning primarily as conditional generation without fully leveraging temporal dependencies in the data. The authors address this limitation by modelling explicitly temporal transitions at each diffusion step (in both the forward and backward processes) through a mixture process.
- **Thorough evaluation**: The authors conduct a strong empirical evaluation using datasets from various domains: scientific spatiotemporal forecasting, video prediction, and time series forecasting. The experiments also include ablations and analyses of key components like dependent noise and the $η$ parameter. In addition to the reported metrics, qualitative visualizations illustrate the strengths of the model espcially compared to regular diffusion.
- **Good presentation**: The paper is well-written, with a logical flow from motivation to results.

**Weaknesses:**

- **Why diffusion models?**: I understand that the main contribution of the paper is extending the capability of diffusion models, and as such it makes sense to focus the evaluation of DyDiff vs DPM. However, I think a general motivation for diffusion/generative models for dynamical data is needed (see questions). Additionally, the authors could discuss how their approach in general relates to previous methods (see questions).

- **Derivations for the reverse process and conditional reverse process are incomplete**: I have a hard time following the derivations in appendix A2 for $q(x_{t-1}^1|x^1_t)$ and $q(x^1_{t-1}|x^1_t, x_0^{-P:1})$, which I find to be too brief. Can the authors explain how $q(x_{t-1}|x_t, x_0^{-P:1})$ is obtained from the forward process?

- **Dataset descriptions lack context**: The Turbulence Flow dataset is introduced without context about what it represents (including what each frame represents physically) or why it's relevant for evaluation. For SEVIR, while mentioned as a "spatiotemporal Earth observation dataset", the significance of Vertically Integrated Liquid (VIL) prediction is not explained. The same goes for the RoboNET, BAIR, and time-series datasets. A few sentences introducing the datasets used are needed in each experimental subsection.

- **The metrics used are not defined**: Critical evaluation metrics (CRPS, CSI, FVD, PSNR, SSIM, LPIPS) are used without proper definition or motivation. This makes interpreting the results difficult. A formal (brief, if needed) definition of these metrics is crucial for the assessment of the empirical evaluation of the paper.

- **Important baselines are missing**: The evaluation is missing important baselines like DYffusion [1], which is mentioned when motivating the present work, but not compared to it empirically. Other more recent relevant work is Rolling Diffusion [2], which the authors may choose to not discuss given how recently it was published.

- **The code is not provided as part of the submission**: while this is optional, I believe it is a useful practice to ensure reproducibility.

**nitpicks**
- could you bold the best numbers in table 8 appendix D2?
- 'Dyffusion' is misspelled in page 2, last paragraph

[1] "DYffusion: A Dynamics-informed Diffusion Model for Spatiotemporal Forecasting", Salva Rühling Cachay, Bo Zhao, Hailey Joren, Rose Yu, NeurIPS 2023

[2] "Rolling Diffusion Models", David Ruhe, Jonathan Heek, Tim Salimans, Emiel Hoogeboom, ICML 2024

**Questions:**

- What are the advantages of using diffusion-like models for dynamical data compared to previous predictive approaches?
- Are there any features of diffusion models that previous methods cannot achieve? can you demonstrare these empirically?
- Can you extend your derivations to show how $q(x_{t-1}|x_0^{-P:s})$ is obtained from the forward process for both the ddpm and ddim sampling methods?

---

> ### Author Response · Authors · 2024-11-20
> **Response to Reviewer hYkb (Part I)**
>
> We sincerely thank Reviewer hYkb for providing insightful reviews and valuable comments. We have addressed your questions in detail below, and the corresponding changes are **highlighted in dark blue** in our revised draft.
>
> **Q1:** Advantages of diffusion models and how DyDiff relates to previous methods.
>
> We outline the advantages of diffusion models as follows:
>
> - **[The Importance of Generative Modeling]** Generative modeling provides a framework for predictive learning, offering the ability to capture complex data distributions. This capability supports high-quality predictions [1], meaningful uncertainty estimation [2], and adaptability to tasks with inherent stochasticity [3-4]. These strengths have enabled generative-based predictive approaches to achieve state-of-the-art performance across diverse modalities and metrics [1-6].
> - **[Diffusion Models as Advanced Generative Models]** Diffusion models have emerged as state-of-the-art generative models, demonstrating superior performance compared to alternatives such as generative adversarial networks (GANs) in recent years [7-8].
>
> Additionally, we have revised the paper to discuss the relationship between our work and previous methods in $\underline{\text{Sec. 5}}$. To summarize, predictive learning methods can be broadly categorized into two paradigms: (1) **auto-regressive approaches** that roll out predictions step-by-step and (2) **all-to-all methods** that predict all targets simultaneously. While **non-generative models** have traditionally dominated both paradigms, there is a growing trend toward leveraging **generative models** due to their superior ability to capture data distributions and improve prediction quality [1,2,3,9]. **Our work focuses on the all-to-all paradigm and employs generative diffusion models.**
>
> [1] *Zhihan Gao et al. Prediff: Precipitation nowcasting with latent diffusion models. In NeurIPS, 2023.*
>
> [2] *Ravuri, Suman, et al. Skilful precipitation nowcasting using deep generative models of radar. In Nature, 2021.*
>
> [3] *Yuchen Zhang et al. Skilful nowcasting of extreme precipitation with NowcastNet. In Nature, 2023.*
>
> [4] *Morteza Mardani et al. Residual corrective diffusion modeling for km-scale atmospheric downscaling. 2024.*
>
> [5] *Vikram Voleti et al. MCVD: Masked conditional video diffusion for prediction, generation, and interpolation. In NeurIPS, 2022.*
>
> [6] *Jaideep Pathak et al. Kilometer-scale convection allowing model emulation using generative diffusion modeling. 2024.*
>
> [7] *Prafulla Dhariwal et al. Diffusion models beat GANs on image synthesis. In NeurIPS, 2021.*
>
> [8] *Jonathan Ho et al. Video diffusion models. In NeurIPS, 2022.*
>
> [9] *Kashif Rasul et al. Autoregressive denoising diffusion models for multivariate probabilistic time series forecasting. In ICML, 2021.*

---

> > ### Comment · Reviewer_hYkb · 2024-11-25
> >
> > P.S.: just a minor detail: could the authors write out the abbreviations for CRPS and CSI the first time they are mentioned in the text (i.e. Section 4.1., paragraph 'Set up').

---

> ### Author Response · Authors · 2024-11-20
> **Response to Reviewer hYkb (Part II)**
>
> **Q2:** Detailed derivations for the reverse process.
>
> We acknowledge that the derivation of the reverse process $q(x_{t-1}^1|x_t^1,x_0^{-P:1})$ may appear concise. **Our proof is based on Eq. (2.115) and Eq. (2.116) in [10]**, which state that, if $p(a)$ and $p(b|a)$ are both Gaussian distributions, then the marginal distribution $p(b)$ and the posterior conditional distribution $p(a|b)$ are also Gaussian.
>
> Below, we expand the proof of the **DDPM-like sampler** using this lemma. The proof consists of two main steps:
>
> - We begin by predefining the conditional Markovian distribution $q(x_t^1|x_{t-1}^1,x_0^{-P:1})$ and aim to prove that the marginal distribution $q(x_t^1|x_0^{-P:1})$ is a Gaussian. The proof is achieved through induction on $t$ from $0$ to $T$. Assume $q(x_t^1|x_0^{-P:1})$ satisfies the required marginal Gaussian distribution, and given $q(x_{t+1}^1|x_t^1,x_0^{-P:1})$ is also a Gaussian, we apply Eq. (2.115) in [10] by letting $a\gets x_t^1,b\gets x_{t+1}^1$, and $c\gets x_0^{-P:1}$ as a global condition,
>
> $$
> \begin{aligned}
> q(x_{t+1}^1|x_0^{-P:1})&=\int_{x_t^1}q(x_t^1|x_0^{-P:1})q(x_{t+1}^1|x_t^1,x_0^{-P:1})dx_t^1 \\\\
> q_c(b)&=\int_{a}q_c(a)q_c(b|a)da
> \end{aligned}
> $$
>
> is also a Gaussian.
>
> - In the next step, it suffices to prove that the reverse process $q(x_{t-1}^1|x_t^1,x_0^{-P:1})$ is a Gaussian. Given that $q(x_t^1|x_{t-1}^1,x_0^{-P:1})$ and $q(x_{t-1}^1|x_0^{-P:1})$ are Gaussian, according to Eq. (2.116) in [10] by letting $a\gets x_{t-1}^1,b\gets x_{t}^1$, and $c\gets x_0^{-P:1}$ as a global condition,
>
> $$
> \begin{aligned}
> q(x_{t-1}^1|x_t^1,x_0^{-P:1})&=\frac{q(x_t^1|x_{t-1}^1,x_0^{-P:0})q(x_{t}^1|x_0^{-P:1})}{q(x_{t-1}^1|x_0^{-P:1})} \\\\
> q_c(a|b)&=\frac{q_c(b|a)q_c(a)}{q_c(b)}
> \end{aligned}
> $$
>
> is also a Gaussian. Therefore, the existence of the reverse process is proved, where the mean and variance could be derived accordingly.
>
> A similar approach applies to the proof of the **DDIM-like sampler**, which we have detailed in the revised draft in $\underline{\text{Appendix A.2}}$.
>
> [10] *Christopher M Bishop. Pattern recognition and machine learning. springer, 2006.*

---

> ### Author Response · Authors · 2024-11-20
> **Response to Reviewer hYkb (Part III)**
>
> **Q3:** Detailed descriptions of datasets and evaluation metrics.
>
> Thanks for your suggestion. We have expanded the descriptions.
>
> - For the datasets, we now provide more details on their key features and the specific task descriptions in $\underline{\text{Sec. 4}}$.
> - For the evaluation metrics, we have added brief introductions in the $\underline{\text{Sec. 4}}$ and provided more detailed explanations in $\underline{\text{Appendix C}}$.
> - Additionally, we have reorganized $\underline{\text{Appendix C}}$ to improve clarity and readability.
>
> **Q4:** Additional baselines, including DYffusion and Rolling Diffusion.
>
> Below, we elaborate on the specific applicability and differences of the suggested baselines:
>
> - **[DYffusion]** While inspired by DYffusion, our Dynamical Diffusion framework is different in its task setup. First, DYffusion is explicitly designed for scenarios with $P=0$ (i.e., only one observed state), as noted in Sec. 2 of [11]: *"we focus on the task of forecasting from a single initial condition."* Second, DYffusion depends on the learnability of an interpolation network over long horizons, which may impose excessive requirements for general prediction tasks, particularly those with shorter horizons or inaccessible intermediate states. In contrast, Dynamical Diffusion is designed for general prediction tasks, applicable to arbitrary observed states.
>
> - **[Rolling Diffusion]** Rolling Diffusion targets temporal dynamics modeling for long videos using a sliding window approach to achieve consistent rolling-out objectives. However, for general prediction tasks, sequence length and rolling-out capability are not the primary concerns. To ensure fairness, we re-implemented the frame-aware timestep approach from Rolling Diffusion for comparison. The table below presents the results on the Turbulence Flow dataset. Despite Rolling Diffusion slightly outperforms standard diffusion models, our Dynamical Diffusion shows the best performance among these methods.
>
> | Method            | CRPS$\downarrow$ ($w8,avg$) | CRPS$\downarrow$ ($w8,max$) | CSI$\uparrow$ ($w5$) |
> | ----------------- | :-------------------------: | :-------------------------: | :------------------: |
> | DPM               |           0.0313            |           0.0364            |        0.8960        |
> | Rolling Diffusion |           0.0311            |           0.0360            |        0.8882        |
> | DyDiff (ours)     |         **0.0275**          |         **0.0315**          |      **0.8998**      |
>
> [11] *Salva Ruhling Cachay et al. Dyffusion: A dynamics-informed diffusion model for spatiotemporal forecasting. In NeurIPS, 2023.*
>
> **Q5:** Regarding the source code.
>
> We have included the source code in this rebuttal. A more polished version will be made publicly available during the camera-ready stage.
>
> **Q6:** Typos and formatting issues.
>
> Thanks for your careful review. The revised draft has corrected the typos and formatting issues you identified.

---

> ### Author Response · Authors · 2024-11-25
> **Request of Reviewer's Attention and Feedback**
>
> Dear Reviewer,
>
> This is a kind reminder that less than two days remain in the 14-day reviewer-author discussion period. We would appreciate your feedback to ensure our response has addressed your concerns.
>
> In response to your suggestions, we have made the following revisions and clarifications:
> - Discussed the **advantages of diffusion models** and clarified how DyDiff **relates to previous methods**.
> - Provided **detailed derivations** for the reverse process to enhance clarity.
> - Added comprehensive **descriptions of datasets and evaluation metrics**.
> - Discussed **additional baselines**, including Dyffusion and Rolling Diffusion.
> - Supplemented the **source code** and addressed **typos and formatting issues**.
>
> Thanks again for your valuable review. We are looking forward to your reply and are happy to answer any future questions.

---

> ### Comment · Reviewer_hYkb · 2024-11-25
>
> Thank you for addressing my questions and concerns. I have adjusted my score accordingly.

---

### Official Review · Reviewer_vb3D · 2024-11-04

**Soundness:** 3
**Presentation:** 3
**Contribution:** 3
**Rating:** 8
**Confidence:** 3

**Summary:**

The authors address temporal learning dynamics - an underexplored area in diffusion models. They introduce Dynamical Diffusion, a simple (essentially one term) modification of existing diffusion dynamics to directly incorporate this temporal dependency in both the forward and backward processes. They show that tracking the temporal similarities results in higher quality samples across multiple modalities.

**Strengths:**

I think this is a good paper. In my view, the motivation is more akin to how momentum methods developed in optimization - in both the cases, tracking the temporal dynamics was important. The idea, in that regard, isn't too novel, but I am glad to see the large number of experiments the authors have performed. I think this will be a paper whose method will be used as benchmark by future papers.

**Weaknesses:**

N/A

**Questions:**

N/A

---

> ### Author Response · Authors · 2024-11-20
> **Response to Reviewer vb3D**
>
> Many thanks to Reviewer vb3D for recognizing our work. We have included the source code with this rebuttal and plan to open-source it during the camera-ready stage, **including the implementation, pre-trained weights, and other relevant resources**.

---

### Official Review · Reviewer_YRPt · 2024-11-04

**Soundness:** 3
**Presentation:** 3
**Contribution:** 2
**Rating:** 6
**Confidence:** 4

**Summary:**

Dynamical Diffusion (DyDiff) is a novel framework that enhances diffusion models for temporal predictive learning tasks. It introduces temporally aware forward and reverse processes that explicitly model temporal transitions at each diffusion step, addressing the challenge of integrating temporal dynamics into diffusion processes. DyDiff achieves this by incorporating a mixture of historical states controlled by a new schedule parameter γ, allowing it to capture temporal dependencies more effectively. The framework is theoretically sound and can be efficiently trained and sampled similar to standard diffusion models. Experiments across scientific spatiotemporal forecasting, video prediction, and time series forecasting demonstrate that DyDiff consistently outperforms standard diffusion models, particularly in generating temporally coherent sequences and improving performance over longer time horizons.

**Strengths:**

The paper introduces a novel framework called Dynamical Diffusion (DyDiff) that incorporates temporal dynamics into diffusion models for predictive learning tasks. This represents an original approach to addressing limitations of existing diffusion-based methods for temporal data. The authors creatively combine ideas from standard diffusion models with explicit modeling of temporal transitions.

The paper is generally well-written and structured logically. Key concepts and the proposed method are explained clearly with helpful illustrations. The authors provide pseudocode for the algorithms, which aids in understanding the implementation. Some technical details are relegated to appendices to maintain flow in the main text.

**Weaknesses:**

1. The paper does not adequately address why temporal dynamics need to be explicitly modeled, given that neural network architectures like transformers can inherently learn such relationships. Specifically:
	-	Lack of justification: The authors do not provide a clear explanation or empirical evidence for why explicitly modeling temporal dynamics is necessary, given that modern architectures like transformers are theoretically capable of learning temporal relationships on their own.
	-	Limited comparison: There is no direct comparison between the proposed Dynamical Diffusion method and transformer-based approaches that implicitly learn temporal dynamics. This makes it difficult to assess the true value added by the explicit modeling of temporal relationships.
	-	Insufficient analysis: The paper does not thoroughly analyze the limitations of existing architectures in capturing temporal dynamics, which would help justify the need for the proposed approach.
	-	Overlooked alternatives: The authors do not discuss potential alternatives for improving temporal modeling within existing frameworks, such as modifications to transformer architectures or attention mechanisms.
	-	Unclear efficiency trade-offs: The paper does not address whether the explicit modeling of temporal dynamics introduces additional computational overhead compared to letting a neural network learn these relationships implicitly.

2. The paper primarily compares Dynamical Diffusion with standard diffusion models. To better establish its significance, the authors should:
- Include comparisons with other state-of-the-art methods in temporal predictive learning, not just diffusion-based approaches.
- Provide a more comprehensive literature review to contextualize their work within the broader field of temporal predictive learning.

3. The paper could benefit from more extensive ablation studies to understand the contribution of different components of the proposed method. Additionally, a thorough analysis of hyperparameter sensitivity would strengthen the work. The authors could:
- Conduct ablation studies on the key components of Dynamical Diffusion.
- Analyze the sensitivity of the method to different hyperparameters, especially the newly introduced γ̄_t schedule.
- Provide guidelines for selecting optimal hyperparameters for different types of temporal data.

**Questions:**

See weakness

---

> ### Author Response · Authors · 2024-11-20
> **Response to Reviewer YRPt (Part I)**
>
> We sincerely thank Reviewer YRPt for providing insightful reviews and valuable comments. We have responded to your questions in detail below, and corresponding changes are **highlighted in green** in our revised draft.
>
> **Q1:** Justification for explicitly modeling temporal dynamics instead of relying solely on implicit deep networks.
>
> We acknowledge that the motivation for explicitly modeling temporal dynamics could have been better elaborated. As a clarification, our proposed Dynamical Diffusion approach is **orthogonal to model architectures and predictive frameworks,** providing a complementary approach rather than a replacement. To further address this point, we have expanded the discussion to include two key aspects:
>
> **Q1.1:** Discussion and comparison with non-diffusion-based methods.
>
> **[Discussion]** We have added a new paragraph in the $\underline{\text{Sec. 5}}$ to discuss existing non-diffusion-based methods. To be specific, predictive learning methods can be broadly categorized into two frameworks: (1) **auto-regressive approaches** that roll out predictions step-by-step and (2) **all-to-all methods** that predict all targets simultaneously. While **non-generative models** have traditionally dominated both paradigms, there is a growing trend toward leveraging **generative models** due to their strong ability to capture data distributions and improve prediction quality. **Our work focuses on the all-to-all paradigm and employs generative diffusion models.**
>
> **[Comparison]** We have already compared non-diffusion-based approaches in $\underline{\text{Table 6-8 (Appendix D) of the original paper}}$ (Table 7-9 of the revised one). These results validate that diffusion-based methods can surpass previous approaches, even those with substantially larger parameters, such as MaskViT and iVideoGPT. However, we also identify scenarios where diffusion-based methods underperform. It is important to note that **our work complements these methods** and advances the field from a different perspective.
>
> **Q1.2:** Discussion on explicitly incorporating temporal relationships within diffusion models.
>
> **[Motivation]** The motivation for this work is similar to research [1-2], which integrates domain-specific physics knowledge into modeling approaches. In predictive learning, the systems we aim to forecast are often governed by continuous temporal dynamics. Imposing such a structure is expected to improve performance.
>
> [1] *Yuchen Zhang et al. Skilful nowcasting of extreme precipitation with NowcastNet. In Nature, 2023.*
>
> [2] *Zhihan Gao et al. Prediff: Precipitation nowcasting with latent diffusion models. In NeurIPS, 2023.*
>
> **[Emperical Advantages]**
>
> - As shown in $\underline{\text{Sec. 4.1-4.3}}$, we conducted comprehensive experiments across various modalities, consistently showing that DyDiff outperforms the standard DPM.
> - Moreover, as illustrated in $\underline{\text{Fig. 4}}$, the standard DPM encounters challenges in separating the foreground object from the background, an issue that is expected to be better addressed by incorporating dynamics. And DyDiff mitigates this problem, as observed in the results. Additionally, analysis in $\underline{\text{Sec. 4.4}}$ reveals that Dynamical Diffusion produces less noisy samples in the early stages of denoising.
>
> **[Compatibility with Advanced Architectures]** In this rebuttal, we conduct additional experiments using Diffusion Transformers (DiT) [3] as the backbone for diffusion models. The table below displays the results. DyDiff significantly outperforms the standard DPM with DiT as well. This result supports that while deep architectures like transformers can capture temporal relationships, **explicitly modeling these dynamics remains beneficial**.
>
> | DiT Backbone  | CRPS$\downarrow$ ($w8,avg$) | CRPS$\downarrow$ ($w8,max$) | CSI$\uparrow$ ($w5$) |
> | ------------- | :-------------------------: | :-------------------------: | :------------------: |
> | DPM           |           0.0434            |           0.0480            |        0.8403        |
> | DyDiff (ours) |         **0.0358**          |         **0.0395**          |      **0.8548**      |
>
> [3] *William Peebles et al. Scalable diffusion models with transformers. In ICCV, 2023.*

---

> ### Author Response · Authors · 2024-11-20
> **Response to Reviewer YRPt (Part II)**
>
> **Q2:** Discussion on efficiency trade-offs.
>
> As outlined in $\underline{\text{Algorithm 1-2}}$, the primary difference between Dynamical Diffusion and standard diffusion models lies in the preparation of inputs and outputs for the denoiser $\epsilon_\theta$. **Importantly, this modification does not introduce any additional forward or backward passes**. As a result, the computational cost of Dynamical Diffusion is **on par with that of standard models**.
>
> - Empirically, in our experiments on the SEVIR dataset, training for one epoch with two RTX 3090 GPUs takes 71 minutes for the standard DPM and 73 minutes for DyDiff.
>
> We have modified our paper to include this discussion in $\underline{\text{Sec. 3.2.2}}$.
>
> **Q3:** Better establishment of the significance of Dynamical Diffusion.
>
> We have followed your suggestions. Please refer to our response to **Q1** for a detailed explanation.
>
> **Q4:** Additional ablation studies.
>
> The design of Dynamical Diffusion involves introducing a dynamic schedule of the variable $\bar{\gamma}_t$ in $\underline{\text{Eq. (3)}}$, along with additional non-independent Gaussian noise, both derived from our design. All other components align with those of standard diffusion models. We have already presented ablation studies on these two key components in the original paper. Below, we provide further explanations to address the reviewer's concerns:
>
> - **[Key components of Dynamical Diffusion]** Without the dynamic schedule $\bar{\gamma}_t$, Dynamical Diffusion reduces to a standard diffusion model. Comparisons are presented in $\underline{\text{Table 1-3 in Sec. 4.1-4.3}}$. In the absence of the additional non-independent Gaussian noise, as illustrated in $\underline{\text{Fig. 7(b)}}$, the model's performance degrades, highlighting the importance of this component.
> - **[Sensitivity of $\eta$]** The sensitivity of $\eta$ has been examined in $\underline{\text{Fig. 7(c)}}$ for the Turbulence Flow dataset. Results indicate that Dynamical Diffusion is relatively robust to variations in $\eta$.
> - **[Guideline for choosing $\eta$]** We provide further analysis by examining different modalities in this rebuttal. Specifically, we conduct experiments on the BAIR dataset. The table below summarizes the results, validating that **DyDiff is robust to the choice of $\eta$ across various modalities**. In our paper, we adopt a straightforward schedule for $\bar{\gamma}_t$ to simplify hyperparameter tuning, requiring only a single additional parameter, $\eta$. While more intricate schedule designs could potentially enhance performance, they also introduce additional complexity. We plan to explore such designs in our future work.
>
> | Method              | FVD$\downarrow$ | PSNR$\uparrow$ | SSIM$\uparrow$ | LPIPS$\downarrow$ |
> | ------------------- | :-------------: | :------------: | :------------: | :---------------: |
> | DPM                 |      48.5       |      25.9      |      92.0      |        4.5        |
> | DyDiff ($\eta=0.1$) |      46.0       |      26.0      |      92.1      |        4.4        |
> | DyDiff ($\eta=0.5$) |    **45.0**     |    **26.2**    |    **92.4**    |      **4.2**      |
> | DyDiff ($\eta=0.9$) |      45.2       |      25.8      |      91.8      |        4.4        |

---

> ### Author Response · Authors · 2024-11-25
> **Request of Reviewer's Attention and Feedback**
>
> Dear Reviewer,
>
> This is a kind reminder that less than two days remain in the 14-day reviewer-author discussion period. We would appreciate your feedback to ensure our response has addressed your concerns.
>
> In response to your suggestions, we have made the following revisions and clarifications:
>
> - Clarified the justification for explicitly modeling temporal dynamics, emphasizing that **Dynamical Diffusion is orthogonal to model architectures and predictive frameworks**.
>   - Added discussions and clarified comparisons with non-diffusion-based methods.
>   - Provided evidence supporting the benefits of explicitly incorporating temporal dynamics.
> - Discussed **efficiency trade-offs**.
> - Better established the **significance of Dynamical Diffusion**.
> - Clarified the **ablation study** and provided **additional sensitivity analyses on other modalities**.
>
> Thanks again for your valuable review. We are looking forward to your reply and are happy to answer any future questions.

---

### Official Review · Reviewer_o9zW · 2024-11-07

**Soundness:** 3
**Presentation:** 2
**Contribution:** 3
**Rating:** 6
**Confidence:** 3

**Summary:**

The authors proposed a new method to approach the problem of predictive learning in the context of diffusion models. Instead of utilizing conditional generation, in which the generation is conditioned on the history, their method devises a forward process that gradually incorporates the history while simultaneously adds noise to the data. In this way, the process mixes the "diffusion time" and the "data time". Training can be done using the usual denoising score matching method.

**Strengths:**

Mixing temporal transitions with diffusion process is a novel idea. The experiments seem extensively done and the proposed method does deliver better performance when compared to DPM in many cases, though not as powerful as heavy models such as iVideoGPT

**Weaknesses:**

1. The writing can at various places uses some improvements. For instance, section 1, from the third paragraph onwards, appear to have large overlap with the abstract. In section 5, why the papers mentioned as "related work" described in the 1st paragraph are related seem to be obscure.
2. The paper does not contain much insight about why iterative scheme like the one described in (4)-(5) is a helpful one, or more precisely, why it is a good structure to impose. Even adding some simple explanations on the basic features of the scheme, such as " larger values of $t$ correspond to a stronger emphasis on historical states", will help bringing some insights.

**Questions:**

1. The author(s) stressed that the method is "theoretically sound" (repeated twice) and "theoretically guaranteed". It'd be good if the author(s) can clarify a. in which sense is it theoretically sound? b. What precisely is "guaranteed" theoretically?
2. I'm confused by the sentence "Notably, the noise factor $\sqrt{1-\alpha_t}$ remains consistent with theoriginal diffusion process along the denoising axis. Hence, the new forward process preserves the same signal-to-noise ratio for any diffusion step $t$". Surely the authors do not mean $\sqrt{1-\alpha_t}$ is independent of $t$, as opposed to what the notation suggests?

---

> ### Author Response · Authors · 2024-11-20
> **Response to Reviewer o9zW**
>
> We sincerely thank Reviewer o9zW for the valuable comments. Below, we provide detailed responses to your questions. The corresponding revisions in the updated draft are **highlighted in red**.
>
> **Q1:** Suggestions to improve writing quality.
>
> We appreciate the reviewer's concern regarding the quality of writing. To address this, we have revised our paper accordingly:
>
> - We have modified the third paragraph in $\underline{\text{Sec. 1}}$ to provide additional details beyond the abstract. Specifically:
>   - Introduced the design of each latent state in Dynamical Diffusion.
>   - Emphasized that the reverse process is defined under theoretical derivation.
> - In the first paragraph of $\underline{\text{Sec. 5}}$, we elaborate on how Dynamical Diffusion connects with existing works, showing that:
>   - "This paper proposes a modification to the diffusion equations, a topic extensively explored in the literature."
>   - "In contrast, our work presents a novel approach that explicitly incorporates temporal dynamics by modifying the diffusion equation."
>
> We have also made revisions to other sections to improve its overall readability.
>
> **Q2:** Insights about the iterative design of Equation (4)-(5).
>
> As outlined in $\underline{\text{Paragraph 1–3 of Sec. 3}}$, our core idea is to extend diffusion models by incorporating predictive axes, which requires defining temporal relationships between latents across timesteps. We opt for the iterative design for the following reasons:
>
> - [**Mathematical Elegance**] The iterative design closely resembles the form of the standard diffusion forward process, enabling us to derive a simplified reverse process. This alignment also facilitates the development of a well-structured algorithm.
> - [**Ease of Implementation**] The resulting formulation is straightforward, simplifying implementation. Furthermore, our method introduces only a single timestep-aware schedule hyperparameter, $\bar{\gamma}_t$, making hyperparameter tuning more manageable.
> - [**Empirical Efficiency**] Iterative designs are widely used in recurrent neural networks [1-2] and state-space models [3], showing their effectiveness in various contexts despite their simplicity.
>
> To address this point, we have revised $\underline{\text{Sec 3.1}}$ of the paper to elaborate on these considerations.
>
> [1] *S Hochreiter. Long short-term memory. 1997.*
>
> [2] *Junyoung Chung et al. Empirical evaluation of gated recurrent neural networks on sequence modeling. 2014.*
>
> [3] *Albert Gu et al. Efficiently modeling long sequences with structured state spaces. In ICLR, 2022.*
>
> **Q3:** Theoretical soundness of Dynamical Diffusion.
>
> In the standard diffusion equations, the forward process is defined as a Markov chain that incrementally adds noise to the data at each timestep, represented by $q(\mathbf{x}^1_t|\mathbf{x}^1_{t-1})$. Through mathematical derivation, the corrupted latent state $q(\mathbf{x}^1_t|\mathbf{x}^1_0)$ at any timestep t can be expressed, where the latter is a result of the former.
>
> In contrast, our approach in $\underline{\text{Eq. (3)}}$ begins by directly defining $q(\mathbf{x}^1_t|\mathbf{x}^1_0)$, motivated by the goal of modeling the predictive axis. This formulation naturally raises theoretical questions, including:
>
> - The existence of a corresponding $q(\mathbf{x}^1_t|\mathbf{x}^1_{t-1})$.
> - The formulation of the reverse process, particularly in multi-step prediction scenarios.
> - Feasibility of optimizing this formulation with practical objectives and performing efficient sampling.
>
> These questions are addressed in $\underline{\text{Sec 3.2}}$ (proofs in $\underline{\text{Appendix A}}$), which supports our claim about the theoretical soundness of the proposed approach.
>
> **Q4:** Clarifying the confusing sentence about the consistency of the noise factor $\sqrt{1-\bar{\alpha}_t}$.
>
> We recognize the confusing sentence. The intended meaning of this sentence is that the choice of noise factor $\sqrt{1-\bar{\alpha}_t}$ is independent of the selection of $\bar{\gamma}_t$. Therefore, we can adopt the same $\sqrt{1-\bar{\alpha}_t}$ schedule in Dynamical Diffusion as standard diffusion models. This ensures $\text{SNR}^{\text{DPM}}_t=\text{SNR}^{\text{DyDiff}}_t$ for all timestep $t$. To improve clarity, we have revised the corresponding sentence.

---

> ### Author Response · Authors · 2024-11-25
> **Request of Reviewer's Attention and Feedback**
>
> Dear Reviewer,
>
> This is a kind reminder that less than two days remain in the 14-day reviewer-author discussion period. We would appreciate your feedback to ensure our response has addressed your concerns.
>
> In response to your suggestions, we have made the following revisions and clarifications:
> - **Rewritten Sec. 1 and expanded Sec. 5** to elaborate on Dynamical Diffusion and its connection to existing works.
> - Clarified the **motivation and soundness of our iterative design**, including mathematical elegance, ease of implementation, and empirical efficiency.
> - Provided a clearer **explanation of the phrase *"theoretically sound/guaranteed."***
> - Charified and **corrected the phrase *"the consistency of noise ratio"***.
>
> Thanks again for your valuable review. We are looking forward to your reply and are happy to answer any future questions.

---

> > ### Comment · Reviewer_YRPt · 2024-11-27
> > **Thanks for the response**
> >
> > I have read the response carefully and don't have any more concerns.

---

> > > ### Author Response · Authors · 2024-11-27
> > > **Invitation to Reevaluate the Rating**
> > >
> > > Thank you once again for your thoughtful reviews. As we have addressed all the concerns raised, we kindly ask if you might consider revising your rating (currently a 5).

---

### Author Response · Authors · 2024-11-20

We would like to sincerely thank all the reviewers for providing insightful reviews and valuable comments. Your reviews are of great importance to us in improving the quality of this work. **We have revised our paper and responded to each reviewer with a separate response.**

---

### Meta-Review · Area_Chair_cWGb · 2024-12-23

**Metareview:**

In this paper the authors introduce Dynamical Diffusion, an extension of the standard diffusion process that renders the process temporally aware. The temporal transitions are modeled explicitly such that overall the temporal dynamics inherent in the data are mixed with the diffusion process.

The paper is well motivated, since the authors seek to fill a gap in the application of diffusion predictive learning, where taking dynamics into account makes sense. Furthermore, the idea is executed well, with the framework remaining scalable through the reparameterization trick.

While the paper can be further improved by comparing to further non-diffusion approaches, in its current state it is a nice well-contained piece of work in the diffusion space, with relevant insights, convincing experiments and text which supports the claims of the paper. In particular, the authors explain that their paper complements other methods and advances the field from a different perspective; I totally agree with this statement and, further, I appreciate the inclusion of scenarios where diffusion-based methods underperform. Overall, a very relevant paper for ICLR.

**Additional Comments On Reviewer Discussion:**

* Various comments regarding improving the writing have been addressed
* Discussion around further comparisons with non-diffusion models ; please see my last paragraph from the meta-review.

---

### Decision · Program_Chairs · 2025-01-22

Accept (Poster)